# Giant magnetoelectric effects achieved by tuning spin cone symmetry in Y-type hexaferrites

Kun Zhai[1,2], Yan Wu [3], Shipeng Shen[1], Wei Tian[3], Huibo Cao[3], Yisheng Chai[1], Bryan C. Chakoumakos[3], Dashan Shang [1], Liqin Yan[1], Fangwei Wang[1] & Young Sun [1,2]

Multiferroics materials, which exhibit coupled magnetic and ferroelectric properties, have attracted tremendous research interest because of their potential in constructing next-generation multifunctional devices. The application of single-phase multiferroics is currently limited by their usually small magnetoelectric effects. Here, we report the realization of giant magnetoelectric effects in a Y-type hexaferrite $Ba_{0.4}Sr_{1.6}Mg_2Fe_{12}O_{22}$ single crystal, which exhibits record-breaking direct and converse magnetoelectric coefficients and a large electric-field-reversed magnetization. We have uncovered the origin of the giant magnetoelectric effects by a systematic study in the $Ba_{2-x}Sr_xMg_2Fe_{12}O_{22}$ family with magnetization, ferroelectricity and neutron diffraction measurements. With the transverse spin cone symmetry restricted to be two-fold, the one-step sharp magnetization reversal is realized and giant magnetoelectric coefficients are achieved. Our study reveals that tuning magnetic symmetry is an effective route to enhance the magnetoelectric effects also in multiferroic hexaferrites.

[1] Beijing National Laboratory for Condensed Matter Physics, Institute of Physics, Chinese Academy of Sciences, Beijing 100190, China. [2] School of Physical Science, University of Chinese Academy of Sciences, Beijing 100190, China. [3] Quantum Condensed Matter Division, Oak Ridge National Laboratory, Oak Ridge, TN 37831, USA. Kun Zhai, Yan Wu and Shipeng Shen contributed equally to this work. Correspondence and requests for materials should be addressed to H.C. (email: caoh@ornl.gov) or to Y.S. (email: youngsun@iphy.ac.cn)

**M**ultiferroics with coexisting ferroelectric (FE) and magnetic orders can realize magnetoelectric (ME) effects due to the cross-coupling between two orders[1–3]. In the past decade, a large number of multiferroic materials and ME devices have been discovered and developed[4–10]. Compared with multiferroic heterostructures and composites, the ME coefficients ($\alpha_H = dP/dH$ or $\alpha_E = \mu_0 dM/dE$, $P$: polarization, $H$: magnetic field, $M$: magnetization and $E$: electric field) of single-phase multiferroics are usually too small to be useful. Therefore, a major challenge for single-phase multiferroics is to explore new mechanisms that will significantly improve the ME coefficients.

Spin-driven multiferroics in which the $P$ is produced via the inverse Dzyaloshinskii–Moriya (DM) interaction[11], spin current model (the KNB model)[12], or exchange striction mechanism[13] with noncollinear or collinear magnetic structures, could yield large ME effects because their FE and magnetic orders are directly correlated. Recently, a number of multiferroics such as $Ni_3TeO_6$ ($\alpha_H = 1300$ psm$^{-1}$)[14], and $Fe_2Mo_3O_8$ ($\alpha_H = 9000$ psm$^{-1}$)[15], high-pressure phase of $GdMnO_3$, $DyMnO_3$ and $TbMnO_3$ ($\alpha_H \sim 3000$–9000 psm$^{-1}$)[16], $GdMn_2O_5$ ($\alpha_H \sim 9000$ psm$^{-1}$)[17] show strong ME effects. However, large magnetic fields (a few teslas) are needed to induce a large change of $P$ in these compounds.

The hexaferrites with tunable conical magnetic structures are among the most promising multiferroics for realizing large ME effects at low magnetic fields. The crystal structure of hexaferrites consists of repeatedly stacking of three structural building blocks: S ($MeFe_4O_8$; spinel block), where Me denotes a divalent metal ion, R [$(Ba,Sr)Fe_6O_{11}$] and T [$(Ba,Sr)_2Fe_8O_{14}$]. The Y-type hexaferrite has an alternating stacking of the S and the T structural blocks along the $c$-axis, as shown in Fig. 1a. Depending on stacking sequences, there are other five types of hexaferrites, including M, W, X, Z and U-type having R

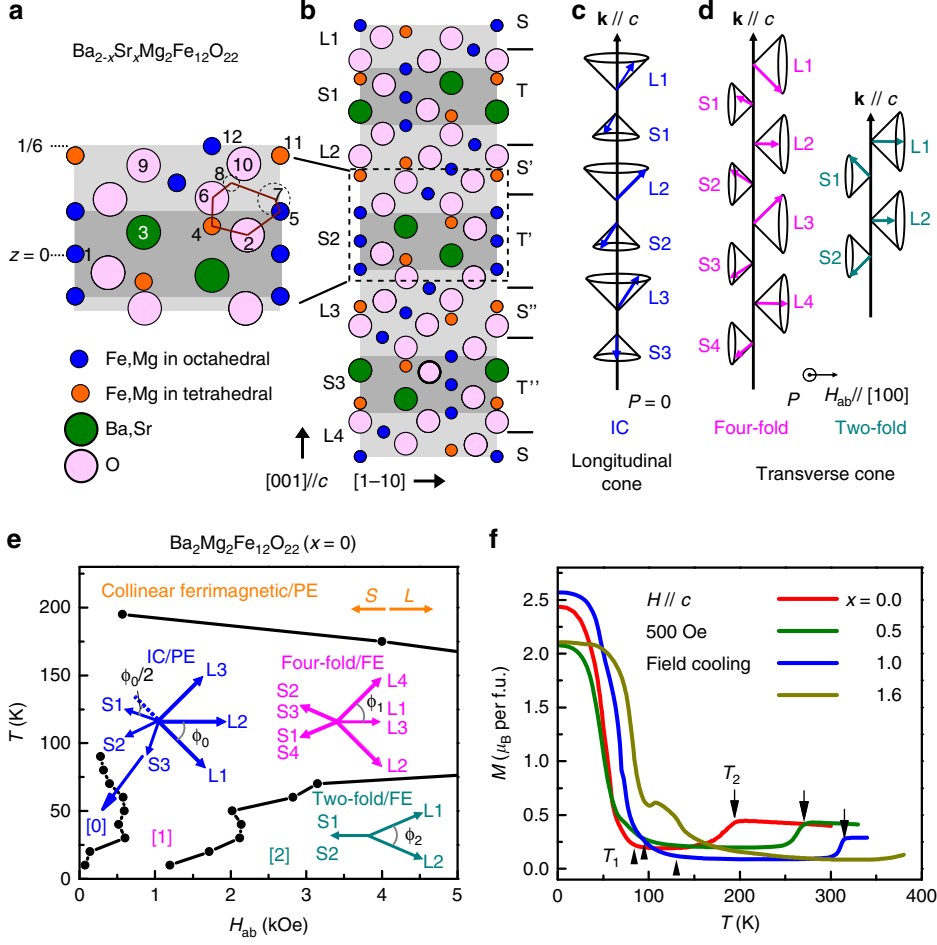

**Fig. 1** Structural and magnetic properties of $Ba_{2-x}Sr_xMgFe_{12}O_{22}$. **a**, **b** Schematic representation of crystal structure of Y-type hexaferrite $Ba_{2-x}Sr_xMg_2Fe_{12}O_{22}$. The dotted square region is magnified in **a**. 1–12 denote the 12 different atomic layers ranging from $z = 0$ to 1/6 in one of ($\mathbf{T}_{1/2}\mathbf{S}_{1/2}$) structure blocks. The *dotted circle* 7 and 8 represent the trajectory of those atoms onto the (110) plane since they lie out of the plane. The connected heavy lines indicate the superexchange paths which would be strongly affected by the Sr doping. The structure blocks are alternating stacked along $c$ as STS'T'S''T''—where the apostrophe means the corresponding block is rotated 120° around this axis. The magnetic structure consists of stacks of L blocks and S blocks along $c$. Illustrations of conical spins (represented by *arrows*) with **c** the incommensurate (IC) longitudinal cone (LC) and **d** the four-fold and two-fold transverse cone (TC). No polarization exists in IC-LC phase while the in-plane $P$ ($\perp H_{ab} \perp k$) can be induced in these TC phases. **e** The magnetic and ME phase diagram of $Ba_2Mg_2Fe_{12}O_{22}$ determined by the measurement of $\varepsilon$-$H_{ab}$. Below 100 K, it is divided into three areas [0], [1], and [2], corresponding to magnetic structures with IC, four-fold and two-fold periodicities. Above 200 K, a collinear ferrimagnetic structure is stabilized. The ferroelectricity (FE) or paraelectricity (PE) for each phase are marked as well. The insets depict the schematic in-plane magnetic structures in each phase from **c** and **d**. $\phi_0$, $\phi_1$ and $\phi_2$ represent the angles between the in-plane moments of neighboring L blocks in IC, four-fold and two-fold phases, respectively. **f** Temperature dependence of magnetization measured after field cooling with 500 Oe magnetic field ($H//c$) of different samples. The *black* and *gray arrows* denote the transition temperatures of LC to proper screw ($T_1$) and proper screw to collinear ferrimagnetic ($T_2$), respectively

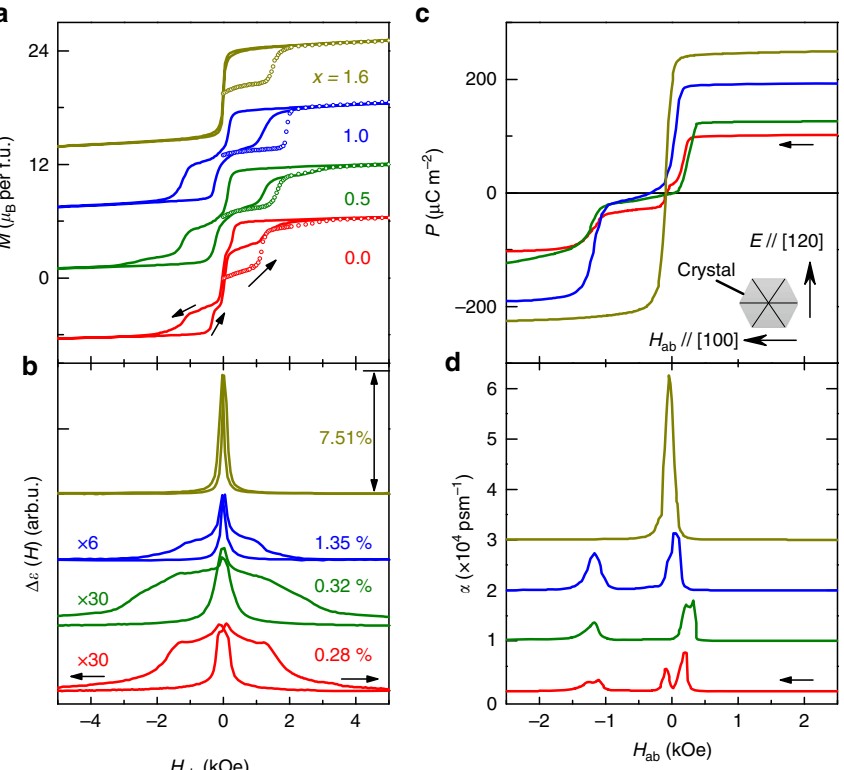

**Fig. 2** Spin-driven multiferroicity in $Ba_{2-x}Sr_xMg_2Fe_{12}O_{22}$ at 10 K. **a** $M$-$H_{ab}$ hysteresis loops for all samples. The initial magnetization curve and normal curves are marked by open circle and line, respectively. **b** The magnetic field dependence of relative dielectric constant $\Delta\varepsilon(H)$ for all samples. $\Delta\varepsilon(H) = [\varepsilon(H)-\varepsilon(5\text{ kOe})]/\varepsilon(5\text{ kOe})$. **c** In-plane electric polarization as a function of $H_{ab}$ for all samples. The inset shows the schematic measurement configuration. **d** The direct ME coefficient $\alpha_H$ as a function of $H_{ab}$ for all samples. The baselines are shifted vertically with constant value for clarity. A single peak of $\alpha_H$ with a giant value of 33,000 $psm^{-1}$ is obtained in $x = 1.6$ sample. The arrows denote the direction of scanning magnetic field

blocks[18, 19]. Kimura et al.[20] first reported $H$-induced FE order in the Y-type hexaferrite $Ba_{0.5}Sr_{1.5}Zn_2Fe_{12}O_{22}$. Later, Ishiwata et al. demonstrated that a low magnetic field of 300 Oe is able to reverse $P$ in the Y-type hexaferrite $Ba_2Mg_2Fe_{12}O_{22}$, indicating a pronounced ME effect[21]. These discoveries have triggered considerable studies on hexaferrites including M-type $Ba(Fe,Sc)_{12}O_{19}$, Z-type $(Ba_{1-x}Sr_x)Co_2Fe_{24}O_{41}$ and U-type $Sr_4Co_2Fe_{36}O_{60}$[22–25]. So far, the largest ME coefficient reported in single-phase multiferroics is from the Y-type hexaferrite $Ba_{0.5}Sr_{1.5}Zn_2(Fe_{0.92}Al_{0.08})_{12}O_{22}$ as $\alpha_H = 20,000$ $psm^{-1}$[26]. Although the multiferroicity and ME coupling in hexaferrites were explained with the spin current model[12] based on noncollinear conical spin structures in the theoretical perspective, it has been elusive what determines the magnitude of the ME effects and how to further enhance the ME effects in hexaferrites.

In this work, we have performed a systematic study on the Y-type hexaferrite $Ba_{2-x}Sr_xMg_2Fe_{12}O_{22}$ family and achieved giant ME coefficients of $\alpha_H = 33,000$ $psm^{-1}$ and $\alpha_E = 32,000$ $psm^{-1}$ in the sample with a nominal composition of $x = 1.6$, both setting a new record in single-phase multiferroics. By exploring how the magnetic phase and ME properties vary with the content of Sr in $Ba_{2-x}Sr_xMg_2Fe_{12}O_{22}$, we are able to establish a clear correlation between the giant ME effects and the symmetry of the conical spin structure.

## Results

### Structural and magnetic properties of the $Ba_{2-x}Sr_xMg_2Fe_{12}O_{22}$ family.
Figure 1a–d shows the schematic crystal and magnetic structures of the Y-type hexaferrites $Ba_{2-x}Sr_xMg_2Fe_{12}O_{22}$ (space group $R\bar{3}m$). According to the relative orientation of

magnetic moments in each layer, two kinds of magnetic blocks: large (L) magnetic moment ($\mu_L$) and small (S) magnetic moment ($\mu_S$) can be defined. The magnetic moments of Fe ions are collinearly arranged within each block[27]. Strong anti-ferromagnetic superexchange interactions cross the boundary of the L and S blocks, especially, Fe(4)-O(2)-Fe(5), Fe(4)-O(2)-Fe(8) and Fe(5)-O(2)-Fe(8) (Fig. 1a). Thus, those competing interactions induce the magnetic frustration at the boundary, and stabilize various noncollinear magnetic structures[28]. Below 100 K, depending on the external in-plane magnetic field ($H_{ab}//[100]$), interactions between magnetic blocks can yield either an incommensurate (IC) longitudinal conical (LC) or a transverse conical (TC) magnetic structure, as shown in Fig. 1c, d, respectively. Ferroelectricity can only be generated in the TC magnetic structures, not LC, via the inverse DM interaction (or spin current model), where $P \sim A \sum_{ij} k \times (\mu_L \times \mu_S)$ ($k$: propagation vector) so that an orthogonal relationship $P \perp H_{ab}$ will hold, as shown in Fig. 1d. Here, $A$ is a scalar determined by the super-exchange interaction and the spin–orbit interaction[12].

The magnetic phase diagram of the parent compound $Ba_2Mg_2Fe_{12}O_{22}$ (BMFO) has been identified in previous studies[29–32]. Figure 1e shows a complex magnetic and ME phase diagram for our BMFO obtained by our systematic magnetodi-electric measurements that are consistent with previous results[29, 32]. Below 100 K and at $H_{ab} = 0$, the system tends to develop an IC-LC magnetic structure which is paraelectric (PE), in which the in-plane magnetic components of $\mu_L$ (or $\mu_S$) periodically rotate around $c$-axis by the turn angle $\phi_0$. Their neighboring L and S always make an angle of $180°-\phi_0/2$ or $180° + \phi_0/2$. At intermediate $H_{ab}$, a four-fold commensurate TC magnetic structure characterized by propagation vector $\mathbf{k} = (0\ 0\ 0.75)$ stabilizes. With further increasing $H_{ab}$, the

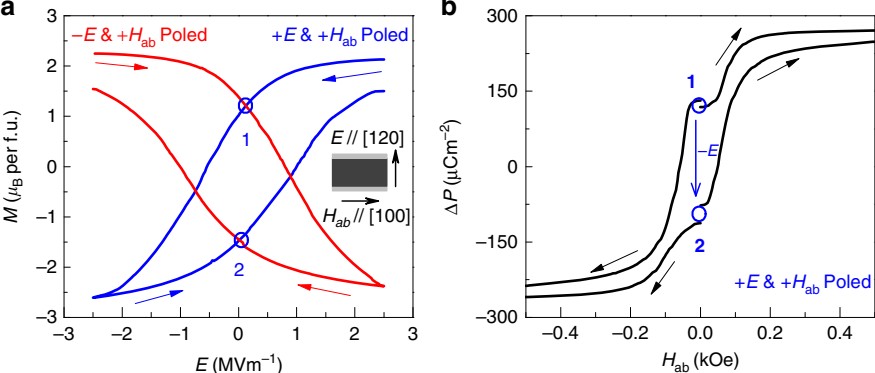

**Fig. 3** *M-E* and *P-H* loops of $x = 1.6$ sample at 10 K. **a** *M-E* loops after $+E$ and $+H_{ab}$ poled or $-E$ and $+H_{ab}$ poled conditions measured at $H_{ab} = 0$. The inset shows the schematic measurement configuration. **b** The $\Delta$ *P-H*$_{ab}$ curves measured from state **1** and state **2** in **a** after $+E$ and $+H$ poled condition

magnetic structure becomes two-fold TC ($\mathbf{k} = (0\ 0\ 1.5)$). Both the four-fold and two-fold TC phases are FE phase. In those phases, the turning angles between neighboring $L$ and $S$ blocks also follow the same relationship. Eventually, the system is driven into the collinear ferrimagnetic (PE) phase with large $H_{ab}$[29]. These consecutive magnetic-field-driven PE-FE phase transitions (IC-LC → four-fold → two-fold or vice versa) occur within a relative large magnetic field scale in BMFO.

Due to the multiple steps of the consecutive phase transition, the ME coefficient $\alpha_H$ is not large[21]. Since the two-fold FE phase possesses the maximum electric polarization[33], the ME effects could be significantly improved by simplifying the ME phase diagram to obtain a one-step LC PE to two-fold FE phase transition. In this work, we substitute Ba with Sr in BMFO to effectively tune the magnetic phase diagram as the four-fold phase is gradually suppressed with Sr substitution.

Firstly, Sr doping changes the lattice constant due to the size difference between Ba and Sr ions. No structure transition is observed within 1.5–450 K. The nuclear structure of the highest doping sample ($x = 1.6$) refined by the neutron diffraction at 5 K is reported in Supplementary Table 1. The lattice constants $a$ and $c$ of the Ba$_{2-x}$Sr$_x$Mg$_2$Fe$_{12}$O$_{22}$ (nominal $x = 0.0$, 0.5, 1.0, and 1.6) samples determined from X-ray diffraction at room temperature decreases systemically with the increase of $x$ (see Supplementary Figs. 1 and 2), implying the systematic doping of Sr[34]. From the single-crystal X-ray diffraction refinement results, the actual Sr/Ba ratio is slightly lower than its nominal value but consistently increasing as does the corresponding nominal Sr level (see Supplementary Table 2).

Secondly, Sr doping also systematically tunes the magnetic transition temperatures. Figure 1f presents the temperature dependence of magnetizations of Ba$_{2-x}$Sr$_x$Mg$_2$Fe$_{12}$O$_{22}$ after field cooling with $H$ ($//c$) = 500 Oe. For $x = 0$, it is a LC phase, which appears as the large spontaneous magnetization along $c$-axis. By increasing the temperature, it enters a proper screw spin structure with $\mathbf{k}//c$ from IC-LC structure above 100 K indicated by the decrease of magnetization ($T_1$). Upon further warming the sample, a collinear ferrimagnetic phase occurs above 200 K ($T_2$)[29]. All the Sr substituted samples keep the same trend in *M-T* curve with a plateau of $M$ below $T_1$ with gradually increased transition $T_1$ at low temperature, indicating the existence of the longitudinal conical state. The $T_2$ also systematically increases with Sr doping, $T_2$ of $x = 1.6$ sample is well above room temperature (420 K). However, the magnitude of the plateau in $M$ below $T_1$ shows no systematic change with doping, which may be contributed to the difference in coercive field. An extra anomaly between 100 and 150 K in $x = 1.6$ indicates a new magnetic phase, different from the proper screw structure. The decreasing of the lattice constants

by Sr doping may lead to the increase of bond angles of Fe-O-Fe, mainly at the boundaries between L and S, which would strengthen the respective exchange interactions[34]. Meanwhile, the magnetic phase transition temperatures are largely elevated.

**Spin-driven ferroelectricity and giant ME effects**. We noticed that applying $H_{ab}$ can drive the LC phase in all Ba$_{2-x}$Sr$_x$Mg$_2$Fe$_{12}$O$_{22}$ samples into the transverse conical state–a noncollinear magnetic order that gives rise to in-plane electric polarization. As shown in Fig. 2a, BMFO ($x = 0$) exhibits a multi-staged $M$-$H_{ab}$ hysteresis loop at 10 K. Three discontinuous magnetization changes are observed at 0.2, 0 and −1.2 kOe during the process of decreasing $H_{ab}$, indicating $H_{ab}$-driven magnetic phase transitions[32, 35]. For $x = 0.5$ and 1.0, there are still two-step magnetic transitions around 0 and −1.2 kOe in the $H_{ab}$ decreasing run. In contrast, for $x = 1.6$, only one abrupt reversal of magnetization occurs at $H_{ab} = -10$ Oe for the $H$-descending branch. These $M$-$H_{ab}$ curves suggest that the $H_{ab}$-driven magnetic phase transitions for Ba$_{2-x}$Sr$_x$Mg$_2$Fe$_{12}$O$_{22}$ are effectively tuned by replacing Ba with Sr. Note that all the virgin magnetization curves lie outside of the subsequent hysteresis loops, implying that the samples could have a different magnetic order after a high-$H_{ab}$ history from what after the zero-field cooling process.

Figure 2b presents the $H_{ab}$ dependence of the relative change of dielectric permittivity $\Delta\varepsilon(H) = [\varepsilon(H) - \varepsilon(5\ \text{kOe})]/\varepsilon(5\ \text{kOe})$ at 10 K in $H_{ab}$ and $E//[120]$. Corresponding to the sharp changes of $M$ in the $M$-$H_{ab}$ curves, $\Delta\varepsilon(H)$ shows clear dielectric peaks at the same magnetic fields. Along with the reduction of dielectric peaks with increasing Sr content, the maximum of $\Delta\varepsilon(H)$ significantly increases, from 0.28% for $x = 0.0$ to 7.51% for $x = 1.6$. This high dielectric peak of $\Delta\varepsilon(H)$ for $x = 1.6$ sample implies a strong ME coupling around zero field.

The ferroelectricity is confirmed by measurements of the ME current at 10 K with $P \perp H_{ab} \perp c$ configuration (see the Methods). As plotted in Fig. 2c, for all samples, $P$ can be fully reversed by applying sufficient $H_{ab}$. For $x = 0$, $P$ exhibits multi-step transitions and the maximum of $P$ is about 103 $\mu$Cm$^{-2}$, which is consistent with the results in ref. [35]. As the Sr content increases, the $P$-$H_{ab}$ curves evolve from multiple steps (three steps for $x = 0$) to one-step reversal of $P$ ($x = 1.6$), which are similar to the $\Delta\varepsilon(H)$-$H_{ab}$ and $M$-$H_{ab}$ curves. Meanwhile, the maximum value of $P$ grows gradually with increasing Sr concentration, from 103 $\mu$Cm$^{-2}$ for $x = 0$–245 $\mu$Cm$^{-2}$ for $x = 1.6$. The enhancement of $P$ by Sr doping can be attributed to the increase of the cone opening angle $\phi_2/2$ in the two-fold TC phase[36] (Figs 1e, 5a) and the enhancement of exchange interactions between L and S blocks, which will be discussed further below.

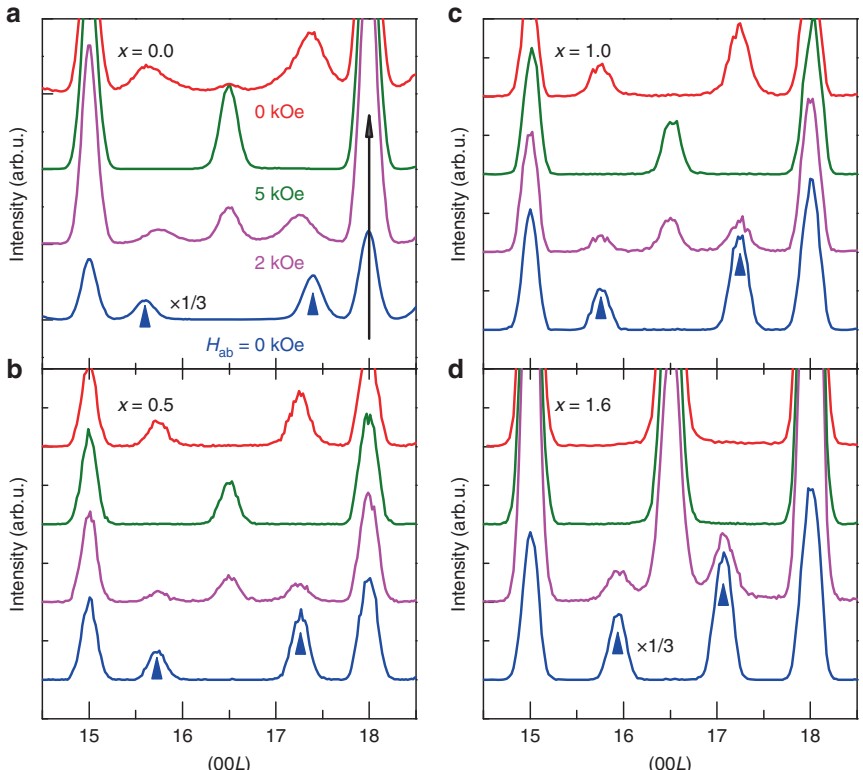

**Fig. 4** Neutron diffraction of $Ba_{2-x}Sr_xMg_2Fe_{12}O_{22}$ at 1.5 K. Neutron diffraction (0 0 L) scan profiles under different $H_{ab}$ for **a** $x = 0.0$, **b** $x = 0.5$, **c** $x = 1.0$ and **d** $x = 1.6$ samples. All the samples initially experienced zero-field cooling down to 1.5 K. Then the magnetic field was changed in the order 0 kOe → 2 kOe → 5 kOe → 0 kOe. (0 0 15) and (0 0 18) peaks are the nuclear diffraction peaks and $L = 3(n \pm \delta)$ denote magnetic diffraction peaks. The arrows indicate magnetic diffraction peaks (0 0 $3n \pm 3\delta_0$) after initial zero-field cooling process

As a consequence, the ME coefficient $\alpha_H$ for $x = 1.6$ reaches as high as 33,000 $psm^{-1}$ around zero-$H_{ab}$ (see Fig. 2d), which is about 1.65 times of the previous record of ME coefficient in single-phase multiferroics (20,000 $psm^{-1}$ in Y-type $Ba_{0.5}Sr_{1.5}Zn_2(Fe_{0.92}Al_{0.08})_{12}O_{22}$ hexaferrite[26]).

**Electric field reversal of magnetization.** Due to the giant direct ME coefficient ($\alpha_H = 33,000$ $psm^{-1}$) and small coercive field, a large converse ME effect is also expected in $x = 1.6$ sample. Especially, the reversal of $M$ direction by applying $E$ field is vital for many applications in spintronic devices but was rarely achieved in single-phase multiferroics[15, 37–39]. To maximize the converse ME effect, we carried out a ME poling procedure on $x = 1.6$ to obtain single $+ M$ & $+ P$ (or $+ M$ & $-P$) compound domain state (see the Methods). Figure 3a shows the $M$-$E$ hysteresis loop of $x = 1.6$ measured in zero magnetic field. The modulation of $M$ by $E$ can reach as much as about 5.3 $\mu_B$ per f.u. with scanning $\pm 2.5$ $MVm^{-1}$ electric field. The converse ME coefficient is obtained by fitting the quadratic equation $\mu_0 M(E,H) = \alpha_E E + \frac{1}{2}\gamma E^2 + \mu_0 M(0,H)$, where $M(E, H)$ is the average of the increasing and decreasing $E$-scan data. The linear term is dominated and $\alpha_E$ is approximated to be $3.2 \times 10^4$ $psm^{-1}$, which is the highest converse ME coefficient in single-phase multiferroics reported so far.

Meanwhile, the reversal of $M$ by $E$ also results in the reversal of $P$ as $P$ and $M$ are coupled in the compound domain state. Therefore, we expect that the states **1** and **2** at $E = 0$ have the opposite remanent $P$ due to their opposite remanent $M$. To confirm our expectation, we performed ME current measurements from states **1** and **2** by sweeping $H_{ab}$ to 5 kOe or $-5$ kOe, where $P$ is saturated, and calculated the $\Delta P$-$H_{ab}$ by integrating ME currents with time. As shown in Fig. 3b, the remanent

$P$ values at state **1** and **2** are opposite in sign with similar magnitude of $\sim$100 $\mu Cm^{-2}$.

**Correlation between spin cone symmetry and the giant ME effects.** The changes of the ME effects in multiferroics are profoundly rooted in the magnetic phase evolution. To illustrate the correlation between the magnetic order and the giant ME coefficients, we performed neutron diffraction on the $Ba_{2-x}Sr_xMg_2Fe_{12}O_{22}$ family to investigate how the noncollinear magnetic structure varies with Sr concentration in external magnetic fields at low temperatures. The nuclear Bragg peaks in the hexagonal structure, $\mathbf{Q}_N$, should satisfy the reflection condition $-h + k + l = 3n$; the rest are purely magnetic[40]. After zero-field cooling process, incommensurate magnetic satellite peaks at $\mathbf{Q} = \mathbf{Q}_N \pm \mathbf{k}_0 = \mathbf{Q}_N \pm 3(0\ 0\ \delta)$ were observed at 1.5 K in zero field for all samples as indicated by the arrows in Fig. 4. For $x = 0$, the modulation wave number $\mathbf{k}_0$ represents the periodicity of in-plane $\mu_L$ (or $\mu_S$) moments along the $c$-axis and the turn angle $\phi_0$ between adjacent in-plane $\mu_L$ (or $\mu_S$) moments in Fig. 1c, e could be calculated accordingly. The $\phi_0$ for $x = 0$ is 71°, which increases to 88° and 91° for $x = 0.5$ and 1.0, respectively. For $x = 1.6$, the incommensurate $\mathbf{k}_0$ becomes (0 0 0.96) and $\phi_0 = 115°$. The systematic changes of $\mathbf{k}_0$ at $H_{ab} = 0$ under the zero-field cooling process indicates that the magnetic ground state of $Ba_{2-x}Sr_xMg_2Fe_{12}O_{22}$ is largely tuned with Sr substitution.

As $H_{ab}$ increases, the incommensurate $\mathbf{k}_0$ peaks gradually disappear and are replaced by the commensurate $\mathbf{k}_1 = (0\ 0\ 0.75)$ (four-fold) or $\mathbf{k}_2 = (0\ 0\ 1.5)$ (two-fold) peaks for all the samples. These are indications of the magnetic phase transition from the longitudinal to the transverse conical magnetic states. For $x = 0$, the four-fold ($\mathbf{k}_1$) and two-fold ($\mathbf{k}_2$) TC magnetic peaks are simultaneously observed at $H_{ab} = 2$ kOe while only the $\mathbf{k}_2$

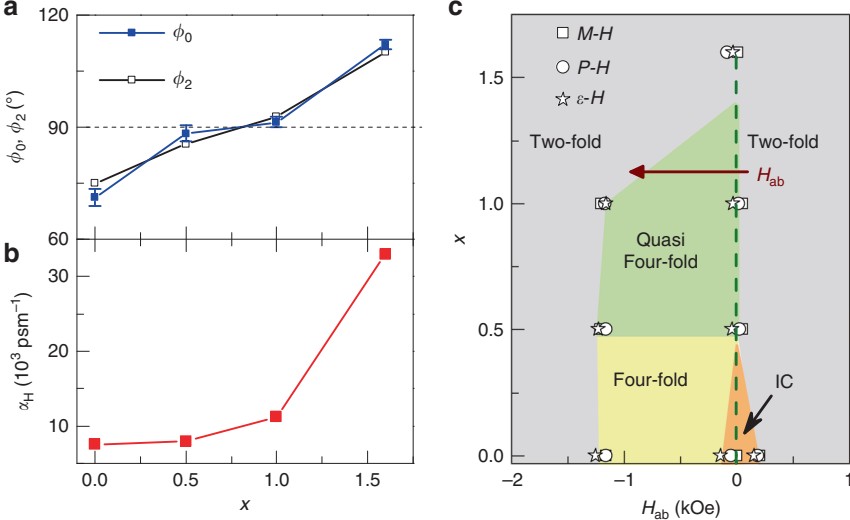

**Fig. 5** Correlation between spin cone symmetry and the ME coefficients. Sr content dependence of **a** the angle $\phi_0$ and $\phi_2$, and **b** the maximum value of $\alpha_H$. $\phi_0$ is determined from the neutron diffraction data with $\phi_0 = 360° \times \delta_0$, where $\delta_0$ is obtained from Fig. 4. The error bar represents the standard deviation of the data. $\alpha_H$ is derived from Fig. 2d. **c** The doping-dependent magnetic phase diagram of $Ba_{2-x}Sr_xMg_2Fe_{12}O_{22}$ in the low $H_{ab}$ region with the down sweep run. The phase boundaries are determined from the M-H, P-H and $\varepsilon$-H curves in Fig. 2, denoted by different symbols

magnetic diffraction peak is observed at $H_{ab} = 5$ kOe. This clearly indicates that BMFO undergoes the transitions of IC-LC to four-fold to two-fold magnetic phases with increasing $H_{ab}$. For $x = 0.5$ and 1.0 samples, the $\mathbf{k}_0$ and $\mathbf{k}_1$ peaks are barely distinguishable within the resolution of our neutron diffraction data with $\mathbf{k}_0 \approx 0.73$ and $\mathbf{k}_0 \approx 0.76$, respectively. Eventually, the samples enter the two-fold TC phase as the 5 kOe data shown. For $x = 1.6$, the sample directly transforms into the two-fold TC phase without showing the $\mathbf{k}_1$ diffraction peak with gradually increasing $H_{ab}$.

To determine the magnetic phase at zero-field after a high-$H_{ab}$ history, we directly swept back $H_{ab}$ from 5 kOe to 0 Oe at the same temperature for each sample. For $x = 0.0$, the incommensurate $\mathbf{k}_0$ peak is recovered mixing with the weak intensity peak of the commensurate component $\mathbf{k}_2 = (0\ 0\ 1.5)$ at zero field. Samples of $x = 0.5$ and 1.0 retain $\mathbf{k}_0/\mathbf{k}_1$ diffraction peaks (note that $\mathbf{k}_0$ and $\mathbf{k}_1$ are indistinguishable in the diffraction peaks for both $x = 0.5$ and 1.0.) without $\mathbf{k}_2$. However, the highest doping sample keeps the two-fold phase till zero field, which implies that the direct reversal of polarization by $H_{ab}$ is attributed to the direct two-fold to two-fold domain transition in the $x = 1.6$ sample. Moreover, by comparing with the M-H data in Fig. 2a, we realize that the detailed initial and final magnetic phases at $H_{ab} = 0$ kOe should be different for $x = 0.5$ and 1.0 samples, although the incommensurate peak positions of those samples return after the removal of external field. The phase differences may be reflected in the slight differences in shape and intensity of the peaks in neutron data.

We summarize the significant change in the turn angle $\phi_0$ with Sr content $x$ obtained from neutron diffraction in Fig. 5a. The threshold of $\phi_0$ to induce four-fold magnetic phase is 90°. For $x = 0.0$, $\phi_0$ is well below 90° in the ground state and a four-fold symmetry can be induced by applying $H_{ab}$. For $x = 0.5$ and 1.0, $\phi_0$ is close to 90° (as shown in Fig. 5a), and the four-fold and incommensurate $\mathbf{k}_0$ diffraction peaks cannot be well separated, which can be regarded as a quasi-four-fold phase. However, the four-fold phase never shows up for the highest doped sample, where $\phi_0$ is much larger than 90°.

Comparing Fig. 5b with Fig. 5a, apparently, there is a close correlation between the turn angle $\phi_0$ and the ME coefficient: as $\phi_0$ increases across 90°, the coefficient $\alpha_H$ rises rapidly. The

complex magnetic phases in the parent compound BMFO can be effectively simplified by changing the $\phi_0$ angle in the ground state. Figure 5c presents the magnetic phases around zero-$H_{ab}$ in the process of decreasing $H_{ab}$ from 5 to −5 kOe for all samples at 10 K. For the low Sr-content samples, the four-fold TC phase or incommensurate LC phase appears during the reversal of P or M. However, for $x = 1.6$, the two-fold TC phase can be stabilized even at zero-field and the reversal of P is just the domain switching of the two-fold TC phase.

## Discussion

Because of the weak in-plane magnetic anisotropy in $Ba_{2-x}Sr_xMg_2Fe_{12}O_{22}$, the angle $\phi_0$ in the ground state is mainly determined by the superexchange interactions across the boundaries between L- and S-blocks, which can be modified by changing the relative content of Ba and Sr. We noticed that the angle $\phi_2$ in two-fold TC also shows a close relationship with the Sr doping, as shown in Fig. 5a. The cone opening angle $\phi_2/2$ or the turn angle $\phi_2$ between adjacent in-plane L moments is estimated from the ratio of the magnetization at high-saturation field and the low field TC phase ($\cos\phi_2/2 = M_{5\ kOe}/M_{50\ kOe}$, see Supplementary Fig. 3). $\phi_2$ increases significantly with increasing Sr doping and has a similar value with that of $\phi_0$ at each doping. The strong $x$ dependency of $\phi_2$ should also be due to the modification of the superexchange interactions by Sr doping. Actually, similar Sr-doping-dependent $\phi_0$ behavior in IC phase has been observed in Y-type $Ba_{2-x}Sr_xZn_2Fe_{12}O_{22}$ and it was attributed to the enhancement of the superexchange interactions across the boundaries between L and S blocks by Sr doping[34]. Moreover, as we discussed above, the increase in $T_2$ value with increasing $x$ is also an experimental proof that the superexchange interaction increases.

To explain the elimination of four-fold symmetry in high Sr doping, we need to study the stabilization condition of each phase with or without magnetic field. According to the previous theoretical calculations, the magnetic Hamiltonian will include the superexchange energies across the L and S blocks and the Zeeman energy $-H_{ab} \cdot M$ if needed. By minimizing the total energies with respect to the turn angles, three in-plane magnetic phases IC, four-fold, two-fold (inset of Fig. 1e) were found to be

stabilized consecutively with increasing $H_{ab}$ where the zero field turn angle $\phi_0$ only depends on the superexchange interactions. Due to the large difference between $\mu_L$ and $\mu_S$, all three phases will have very close superexchange energy terms, leading to three almost identical turn angles in the low $H_{ab}$ region. Such angle constraints are predicted and observed in similar Y-type $Ba_{2-x}Sr_xZn_2Fe_{12}O_{22}$ ($x = 1.5$)[34, 41], which seems to be true in our system as well, with $\phi_0 \approx \phi_2$ for every doping at low $H_{ab}$. On the other hand, the sequence of the phases under increasing $H_{ab}$ is governed by their net $M$. In other words, phases with larger $M$ will tend to be stabilized at higher field due to their lower Zeeman energy.

Then, we can understand the missing of the four-fold symmetry in high Sr doping in our system from the above discussions. In the four-fold phase of Y-type $Ba_{2-x}Sr_xZn_2Fe_{12}O_{22}$, all the spins are in the $ab$-plane so that L1 to L4 will have the same in-plane moments with L1 and L3 paralleling the $H_{ab}$. This phase will allow a large net $M$ even when $\phi_1 > 90°$. In contrast, the four-fold phase of our system is a transverse cone spin configuration, where L1 to L4 tilt away from $H_{ab}$ and lead to a much smaller net $M$ for $\phi_1$ close to or larger than 90°, as shown in the inset of Fig. 1e. This is indeed the case for high Sr doping with $x > 1.5$ in two systems where all the turn angles are larger than 90°. We can compare the Zeeman energy $-H_{ab} \cdot \sum(\mu_L - \mu_S)$ between four-fold TC in $Ba_{0.4}Sr_{1.6}Mg_2Fe_{12}O_{22}$ with four-fold phase in $Ba_{0.5}Sr_{1.5}Zn_2Fe_{12}O_{22}$. The former cone phase can be energetically lifted by external $H_{ab}$. On the other hand, the Zeeman energy of two-fold phase in both systems are energetically nearly the same since the magnetic moment of L blocks in two-fold TC phase are completely in the plane (Fig. 1d and inset of Fig. 1e). As a result, the complex phase diagram of $Ba_2Mg_2Fe_{12}O_{22}$ can be greatly simplified only in the high Sr-doped sample ($\phi_0 > 90°$).

The simplification of the phase diagram for the $x = 1.6$ sample leads to the direct reversal of $P$ by $H$ and giant ME effect. Besides, the enhancement of $P$ values by Sr doping is another factor to achieve the giant ME coefficient. Firstly, the increase of cone angle opening $\phi_2$ can enhance the $P$. According to ref. [36], the polarization value $P \sim A\mu_S\mu_L\sin^2(\phi_2/2)$ for the two-fold TC model, where $A$ is determined by exchange interaction and spin–orbital coupling. From this equation, the estimated $P$ at 5 kOe for $x = 1.6$ is 1.82 times larger than that for $x = 0$ in the two-fold phase if $A$ is a constant. Secondly, the change of DM interaction $A$ by Sr doping must have a role. We found that $\mu_S$ and $\mu_L$ are almost invariant due to the similar saturation magnetization at 50 kOe for all doping (Supplementary Fig. 3). Then the extra enhancement of $P$ comes from Sr doping enhanced $A$. In fact, in Y-type hexaferrite, the inverse DM mechanism induced $P$ is mainly generated by the noncollinear spins at the boundary region between L and S, especially, the Fe(4), Fe(5) and Fe(8) layers[40] which is highly frustrated and easily influence by Ba/Sr doping. As we show above, the bond angles and superexchange interactions between those layers are sensitive to the Sr doping. Therefore, the strength of the inverse DM interaction $A$, which is proportional to the superexchange interactions[12], is enhanced by Sr doping. This explains the extra increase of $P$ value. With the effect of two aspects above, a giant ME coefficient is achieved in the $Ba_{0.4}Sr_{1.6}Mg_2Fe_{12}O_{22}$ with $\alpha_H = 33,000$ psm$^{-1}$.

In summary, we have achieved unprecedentedly high ME coefficients in $Ba_{0.4}Sr_{1.6}Mg_2Fe_{12}O_{22}$, where noncollinear conical spin structures produce ferroelectricity through the spin current model. A systematic study on the $Ba_{2-x}Sr_xMg_2Fe_{12}O_{22}$ family reveals that the symmetry of the conical spin structure plays a critical role in improving the ME effects. When the spin cone symmetry was constrained to be two-fold, a one-step sharp

magnetic transition is achieved around zero magnetic field, leading to a giant ME coefficient. Realizing the giant ME coefficients through the spin cone symmetry tuning in this work suggests a route to enhance the ME effects in multiferroic hexaferrites: condensing the interacting magnetic/FE blocks in a smaller unit likely directs a stronger ME coupling and larger magnetic/FE polarization.

## Methods

**Sample preparation and characterization**. Single crystals of $Ba_{2-x}Sr_xMg_2Fe_{12}O_{22}$ with nominal $x = 0.0, 0.5, 1.0, 1.6$ were prepared by crystallization from high-temperature $Na_2O$-$Fe_2O_3$ flux melted at 1420 °C in Pt crucible and slowly cooled to 1100 °C after a thermal recycle[42]. The as-grown crystals were characterized by X-ray diffraction at room temperature (see Supplementary Fig. 1). From the single-crystal X-ray diffraction refinement results, the actual Sr/Ba ratio of each sample is determined, which is slightly lower but consistently increasing as its corresponding nominal Sr level (see Supplementary Table 2 and discussions). The structure of the highest doping sample ($x = 1.6$) was measured at 5 K by neutron diffraction (see Supplementary Table 1).

**Dielectric and ME current measurements**. The dielectric and ME current measurements were carried out in a Cryogen-free Superconducting Magnet System (Oxford Instruments, Teslatron PT) using a LCR meter (Aglient 4980 A) and a high-resistance electrometer (Keithley Model 6517B) respectively. The samples were polished as a thin plate (2 mm × 2.5 mm × 0.085 mm) and electrodes contacts were made of silver pastes painted on the largest surfaces. $H$ was applied along [100] direction and $E$ was applied along [120] direction. The polarization is obtained by integrating the ME current with time. Before ME current measurements, a ME annealing procedure was applied as following: an external magnetic field was set to 50 kOe then ramp to 5 kOe to cross the PE to FE phase boundary, with $E = 750$ kV m$^{-1}$ applied to the samples to align the FE domains. $E$ was switched off and electrodes were shorted for 30 min before sweeping $H$ to obtain ME current.

**Magnetization measurements**. Magnetization measurements were carried out in a Magnetic Properties Measurement System (MPMS-XL, Quantum Design). Using a modified sample holder enabling the application of $E$, $M$ can be measured under electric field. The modulation of magnetization is measured after identical ME annealing condition with both a 50 kOe $H$ along [100] and $\pm$ 2.5 MV m$^{-1}$ $E$ along [120]. Magnetic field was swept back to 0 Oe to measure the $M$-$E$ loops.

**Neutron diffraction measurements**. Single-crystal neutron diffraction experiments were carried out on the fixed-incident-energy triple axis spectrometer HB-1A and HB-3A four-circle diffractometer at the High Flux Isotope Reactor, Oak Ridge National laboratory. The typical size of the crystals is in the range of 3 mm × 3 mm. At HB-1A, a monochromic neutron beam of wavelength $\lambda = 2.358$ Å ($E_i = 14.6$ meV) was employed through a double pyrolytic graphite monochromator system. Highly oriented pyrolytic graphic filters were placed after each monochromator to significantly reduce the higher-order contaminations of the incident beam The experiment was performed using a vertical field magnet with the crystal oriented in the ($H$ 0 $L$) scattering plane. Data were collected at fixed temperature of $T = 1.5$ K with $H$ // [100]. The measurement at HB-3A used the neutron wavelength of 1.546 Å from a bent Si-220 monochromator. The data were refined with FullProf[43].

**Data availability**. The data that support the findings of this study are available from the corresponding authors upon reasonable request.

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

## Acknowledgements

This work was supported by the National Key Research and Development Program of China (Grant No. 2016YFA0300700) and the National Natural Science Foundation of China (Grant Nos. 11534015, 11374347 and 11675255). Y.S. also acknowledges the support from Chinese Academy of Sciences (Grants No. XDB07030200 and KJZD-EW-M05). The work (Y.W., W.T. and H.C.) at ORNL's High Flux Isotope Reactor was sponsored by the Scientific User Facilities Division, Office of Basic Energy Sciences, the US Department of Energy.

## Author contributions

Y.S. and H.C. supervised this study. K.Z. prepared the samples and carried out dielectric and magnetic measurements with the help of S.S. Y.W., W.T. and H.C. performed neutron diffraction measurements. Y.W., H.C. and B.C.C. performed X-ray diffraction measurements. Y.C., D.S., L.Y., F.W. contributed to the data analysis and discussions. Y.S., K.Z. and S.S. wrote the paper, and all authors reviewed the paper.

## Additional information

**Competing interests:** The authors declare no competing financial interests.

