## [Peer Review File · Nature Communications]

Reviewers' comments:

Reviewer #1 (Remarks to the Author):

The authors reported realization of the largest magnetoelectric coefficient in the single-phase multiferroic materials. Their target material is a series of Sr-doped $\text{Ba}_2\text{Mg}_2\text{Fe}_{12}\text{O}_{22}$ where the non-doped compound is one of the most famous spin-spiral multiferroics. In $\text{Ba}_2\text{Mg}_2\text{Fe}_{12}\text{O}_{22}$, the two-fold (so-called FE3) phase with the largest P and M is reached upon application of H via intermediate four-fold (FE2') phase which has smaller P and M. Although the FE2' phase becomes metastable at H=0 for x=0, due to this separated two-step transition nature, ME coefficient measured upon M-reversal (which is accompanied by P-reversal) was smaller. The authors found that one can destabilize FE2' phase by tuning interblock exchange coupling through chemical substitution. As a result, FE3 with large P and M is stabilized as a metastable state even at H=0, which maximize ME coefficient upon reversal of M. Its inverse effect, namely E-induced reversal of M is also confirmed and change in magnetization reaches 5.3 $\mu\text{B}/\text{f.u.}$ Apart from the quite low operation temperature (10 K), these are the world records of the ME coefficients in the single phase multiferroic materials and thus, in my opinion, will attract broad and vital interest of researches. Therefore I would like to recommend this paper for publication in Nature Communications if the authors consider the following questions and comments.

Questions and comments:

1. The authors tried to give a qualitative explanation why four-fold state is destabilized for x=1.6. Their explanation is based on the evolution of angle ϕ in the ground states upon doping Sr. For me, it is not straightforward. Why incommensurate k_0 should be larger than commensurate k_1 and k_2 (see p11)? For x=1.0, $k_0=0.59$ and $k_1=0.75$. In addition, $k_0=0.96$ and $k_2=1.5$ for x=1.6. Therefore, this condition is not fulfilled. In addition, as the authors suggested, ϕ is 115 degs for x=1.6 and this is off from 90 degs. But 115 degs is still closer to 90 degs rather than 180 degs. Then, why the four-fold phase is destabilized compared to the two-fold phase? Also, in the authors' discussion, only L-blocks are considered. Why the S- blocks can be neglected to discuss the stability of each phase? Is it possible to estimate angles between μ_L and μ_S in each of phases from magnetization and discuss relative stability between two-fold and four-fold phases? Are there any changes in magnetic anisotropy upon doping Sr and can it be neglected to discuss stability of these magnetic phases? Whatever the case, more straightforward explanation is necessary.
2. Related to above mentioned comments, the authors pointed out that the angle between neighboring μ_L and μ_S changes in two-fold magnetic phase upon increasing x so that P gets larger. Are there any proofs for this change in angles other than change in P? For example, change in magnetization in two-fold phase upon doping Sr will be useful to estimate such changes in angles and thus should be discussed. P for x=1.6 is twice as large as that for x= 0.5 in two-fold phase, while the magnetization values in two-fold phase in these two compounds are not so different. In my opinion, the difference in P is too large to be ascribed to the difference in the angle between neighboring μ_L and μ_S . Is it really reasonable?
3. It is impossible to judge whether E-induced M-reversal is accompanied by P-reversal or not only from the M-E curve. Therefore, P-E curve for x=1.6 should also be provided, if possible.
4. M-T curve for x=1.6 (Fig.1e) shows anomaly around 100-130 K. What is the origin of this?
5. In line 8, page 35, references 33-35 are cited for the previous works on E-induced reversal of M. However, ref. 33 does not include such results at all. Instead, ref. 29 should be cited here.
6. Samples measured here are grown by the flux method. Then, the composition of grown crystal, especially the x value can be different from the starting compositions. How do the authors determine x value for each composition?

7. In page 13 "The magnetic properties were carried out..." should be replaced by, for example "Magnetization measurements were carried out..."
8. In the method section, typical dimensions of the samples for each measurement should be provided.

Reviewer #2 (Remarks to the Author):

Manuscript NCOMMS-17-00859

Giant magnetoelectric effects achieved by tuning spin cone symmetry in Y-type hexaferrites by Kun Zhai et al.

The Authors claim that "This work discloses an effective route to enhance the magnetoelectric effects by tuning magnetic symmetry." In the paper the way to tune magnetic properties of multiferroic $\text{Sr}_2\text{Mg}_2\text{Fe}_{12}\text{O}_{22}$ is a change of its chemical composition e.g. substitute part of Sr ions by Ba ions. Multiferroic materials exhibit coupled magnetic and ferroelectric properties. Therefore the method of changing their magnetic properties and structures with their chemical composition is a common effect observed and reported in many papers. Composition changes, were widely used in ABO_3 multiferroics to improve their magnetoelectric properties. It is widely accepted that the Dzyaloshinskii-Moriya (D-M) interaction is sensitive to chemical composition of the compound. Therefore the effect presented in the paper was already known. In multiferroics atomic composition change causes a change of their crystal and magnetic structure. This is a consequence also of the Dzyaloshinskii-Moriya interaction present in these materials. Therefore the last sentence in the abstract (lines 31 and 32) "This work discloses an effective route to enhance the magnetoelectric effects by tuning magnetic symmetry" suggests that this tuning was shown the first time by the Authors. I propose to withdraw this sentence from the abstract, or precisely describe what a new idea of improving the magnetoelectric properties of electromagnets is presented in the paper.

As hexaferrites are paraelectric in the absence of external magnetic fields what limits their application in a technique. The single crystal of hexaferrite, $\text{Ba}_{0.4}\text{Sr}_{1.6}\text{Mg}_2\text{Fe}_{12}\text{O}_{22}$, investigated in the paper shows a big magnetoelectric coefficient and a large electric-field-reversed magnetization $\Delta M = 5.3 \mu\text{B/f.u.}$ It is very interesting and perspective result. This result justifying publication of the paper after its solid reconstructions.

The Authors rapport the changes of magnetic structure of compounds as a function of x (Ba/Sr) ratio via neutron scattering and using magnetization, ferroelectric, and neutron diffraction measurements (for four solid solutions).

Line 61-66 "The study on the Y-type hexaferrite $\text{Ba}_{2-x}\text{Sr}_x\text{Mg}_2\text{Fe}_{12}\text{O}_{22}$ family and achieved giant ME coefficients of $d_H = 33000 \text{ psm}^{-1}$ and $d_E = 32000 \text{ psm}^{-1}$ in $\text{Ba}_{0.4}\text{Sr}_{1.6}\text{Mg}_2\text{Fe}_{12}\text{O}_{22}$, is a new record in single-phase multiferroics". By exploring how the magnetic phase and ME properties vary with the content of Sr in $\text{Ba}_{2-x}\text{Sr}_x\text{Mg}_2\text{Fe}_{12}\text{O}_{22}$, the Authors claim that they observe the correlation between the giant ME effects and the conical spin structure. This fact has to be better described and documented (see below). It has to be compared with papers of the results of other groups working with these class of materials (e.g. reference [30]).

Paper needs with following changes:

- Withdraw last sentence in the abstract,

- In the Introduction refer to the papers investigating similar effects in other multiferroics,
- D-M and magnetic interactions should be described for Y-type hexaferrites.
- The crystal and magnetic structures and their changes for different Ba/Sr ratio has to be presented in the paper (values and their errors). For at least one solid solution e.g. for Ba_{0.4}Sr_{1.6}Mg₂Fe₁₂O₂₂ the complete structure analysis (also reliability factors R) has to be presented.

Reviewer #3 (Remarks to the Author):

The present manuscript focusses on Sr-doping of the Y-type hexaferrite Ba₂Mg₂Fe₁₂O₂₂ (BFMO) and on its effects on its dielectric and magneto-electric properties. There is a vast amount of literature on Y-type hexaferrite with various cation compositions and doping, but the general understanding of the properties of these materials is poor, partly due to the complexity of the magnetic structures at the microscopic level (beyond the 'block' model). Unquestionably, this paper provides an original contribution to the field, particularly in that the Sr/Ba substitution, although a very natural one, has not been explored in detail. The paper contains much valuable information about the phase diagram, which will be of interest for the researchers that are focussing on these compounds. In this report, I therefore focus on the elements that would make this paper publishable with the high degree of priority and exposure to a wider audience that is granted by Nature Comm. Based on the abstract, this claim is largely based on the giant magneto electric effect and on the large electric-field-reversed magnetisation.

1. Giant magneto-electric effect: there is a plethora of 'giant' effects at low temperatures (e.g., 'Colossal magnetoresistance' in some manganites), and very few of them are of any technological interest whatever. These effects would, of course, be interesting if they denoted new physics, but this does not seem to be the case here. The so-called 'giant magneto-electric effect' here is simply a consequence of the fact that the transverse conical phase exists in zero field for $x = 1.6$. This cone is very easily manipulated so one sees a large change in P (equal to $2P_{max}$) over a narrow magnetic field interval - hence a large ME effect. In other words, the $x = 1.6$ is simply a continuation of the phase existing at high doping, but 'compressed' towards the zero-field axis. It is not even clear that this phase even displays the linear magneto-electric effect by symmetry,
2. The electric-field-reversed magnetisation is, I think more interesting, but the explanation of why this effect occurs is extremely confusing. In general, at these low temperatures there is a huge amount of irreversibility, with many phases coexisting due to either metastability or kinetic trapping (this is clearly visible in the M-H loops). Under these circumstances, it is extremely difficult to understand what is going on, and it may not even be clear which phases are truly polar.

The paper is not particularly well written, and the non-specialist reader would clearly be confused by the continuous references to previous findings (some of them rather dubious) in a very specialised subfield. I have no doubt that the paper could be greatly improved, but I am quite convinced that no improvement could raise to the required standard. I therefore recommend against publication in Nature Comm.

Reviewer #4 (Remarks to the Author):

Manuscript: NCOMMS-17-00859

Title: Giant magnetoelectric effects achieved by tuning spin cone symmetry in Y-type hexaferrites

The authors have presented an intriguing result in regards to extremely large magnetoelectric effects

in the family of Y-type hexaferrite $\text{Ba}_{2-x}\text{Sr}_x\text{Mg}_2\text{Fe}_{12}\text{O}_{22}$. By studying a range of Sr doping levels, and utilizing a number of characterization techniques, the authors are able to develop a model explaining the large effect observed for $x=1.6$. While an intriguing result worthy of publication, the authors need to address a number of issues in regards to the presentation of their results before they will be suitable for publication. These issues are listed below.

1) Structure of the Y-type hexaferrite $\text{Ba}_{2-x}\text{Sr}_x\text{Mg}_2\text{Fe}_{12}\text{O}_{22}$

For the general reader, the authors should provide a bit more information on the structure of these hexaferrites, for example an explanation of the notation of Y-type, vs. M-, Z-, and U-types. In the crystal structure schematic in Fig. 1(a), the authors should include labels for the in-plane and out-of-plane directions. The yellow atoms may not reproduce well. The authors should give an explanation of the notation of Fe(5) and Fe(4) used in Fig. 1(a). From their figure, it is not possible to visualize the octahedral and tetrahedral coordination of the $\text{Fe}^{3+}/\text{Mg}^{2+}$ ions. Is that an important factor in this work? What determines the magnitude of the Large and Small magnetic moment layers? On line 102 on the top of page 6, it is not clear what is meant by “calibrating” the neutron and x-ray diffraction data. The two measurements are taken at different temperatures, so how are thermal expansion coefficients taken into account? Are there any structural transitions that occur as a function of temperature between room temperature and 5 K that would need to be taken into account?

2) Magnetoelectric phase diagram of $\text{Ba}_2\text{Mg}_2\text{Fe}_{12}\text{O}_{22}$

In Fig. 1(d), the authors present their data on the magnetoelectric phase diagram of $\text{Ba}_2\text{Mg}_2\text{Fe}_{12}\text{O}_{22}$ and denote three regions where the material exists in different magnetic structures. It is not clear whether the data points in green, red, and blue represent experimental points which delineate the transition between these three regimes, or are these data points within each regime? The insets within the panel are a bit small and difficult to read. What is represented by the red arrow?

3) Magnetic properties of Y-type hexaferrite $\text{Ba}_{2-x}\text{Sr}_x\text{Mg}_2\text{Fe}_{12}\text{O}_{22}$

The authors present the M vs T curves of their samples in Fig. 1(e). On line 105 on the top of page 6, the authors should highlight that the transition into the proper screw spin structure corresponds to the decrease in magnetization seen below 195 K. The authors neglect to discuss the composition dependence of several features in their M vs T curves, for example, the transition temperature into the proper screw spin structure, as well as the transition temperature into the LC state, and the value of the plateau in magnetization reached at low temperature. Furthermore, the shape of the M vs T curves differs in the $x=1.6$ sample compared to the other compositions. What is the significance/origin of this feature which appears between 100 and 150 K? The authors should indicate what is meant by the FC notation in Fig. 1(e). No y-axis values are listed for Fig. 2(a). An explanation should be given for the lack of values, or the same units as used in Fig. 1(e) and Fig. 3 should be used. The figure caption to Fig. 2 should explain that the dotted line represents the initial magnetization curve. It is noted, however, that this curve is difficult to distinguish from the other curves in the plot. The authors state that the hysteresis loops shown in Fig. 2(a) have three discontinuous transitions for the $x=0$ composition, however, this statement ignores the gradual increase in magnetization which occurs for $H < -1.2$ kOe, such that the material does not reach full saturation until above -2 kOe. This feature is also prevalent in the $x=0.5$ sample, though it seems to disappear for $x=1$ and 1.6 . Some discussion should be made for these features. Do the sharp transitions seen in Fig. 2(a) correspond well to the field values of the transitions Fig. 2(b)? The authors should give an explanation of why the H and E fields are applied along the given crystallographic directions. The authors should add a reference to Fig. 2(d) in the text at the bottom of page 7.

4) Overall paper structure

The authors have not maintained a consistent choice of color to represent a given sample throughout all the figures in the manuscript. For example, in Fig. 1(e), the colors are black, red, blue, and pink for the $x=0, 0.5, 1.0,$ and 1.6 samples, respectively, while the colors are red, green, blue, and dark yellow for Fig. 2. A different color scheme is used in the supplemental material. The composition of BMFO is given correctly on line 227 on page 11. The authors should give some actual dimensions of the samples in the methods section, since the sample dimensions affect the voltage values needed to

achieve a given E field. On line 280 on page 14, "monochromatic" is misspelled. On line 285 of page 14, the Miller index notation is given using the notation for a direction, rather than a plane.

Responses to the reviewers' comments

Reviewer #1 (Remarks to the Author):

The authors reported realization of the largest magnetoelectric coefficient in the single-phase multiferroic materials. Their target material is a series of Sr-doped $\text{Ba}_2\text{Mg}_2\text{Fe}_{12}\text{O}_{22}$ where the non-doped compound is one of the most famous spin-spiral multiferroics. In $\text{Ba}_2\text{Mg}_2\text{Fe}_{12}\text{O}_{22}$, the two-fold (so-called FE3) phase with the largest P and M is reached upon application of H via intermediate four-fold (FE2') phase which has smaller P and M. Although the FE2' phase becomes metastable at H=0 for x=0, due to this separated two-step transition nature, ME coefficient measured upon M-reversal (which is accompanied by P-reversal) was smaller. The authors found that one can destabilize FE2' phase by tuning interblock exchange coupling through chemical substitution. As a result, FE3 with large P and M is stabilized as a metastable state even at H=0, which maximize ME coefficient upon reversal of M. Its inverse effect, namely E-induced reversal of M is also confirmed and change in magnetization reaches 5.3 uB/f.u. Apart from the quite low operation temperature (10 K), these are the world records of the ME coefficients in the single phase multiferroic materials and thus, in my opinion, will attract broad and vital interest of researches. Therefore I would like to recommend this paper for publication in Nature Communications if the authors consider the following questions and comments.

We appreciate the reviewer's efforts in carefully reviewing our manuscript and agreeing the great importance of our work for publication in Nature Communications. Apparently, this reviewer is an expert in multiferroic hexaferrites, and his/her instructive comments have helped us to make a significant improvement of our paper. The responses to all the questions and comments are listed below.

Questions and comments:

1. The authors tried to give a qualitative explanation why four-fold state is destabilized for x=1.6. Their explanation is based on the evolution of angle phi in the ground states upon doping Sr. For me, it is not straightforward. Why incommensurate k_0 should be larger than commensurate k_1 and k_2 (see p11)? For x=1.0, $k_0=0.59$ and $k_1=0.75$. In addition, $k_0=0.96$ and $k_2=1.5$ for x=1.6. Therefore, this condition is not fulfilled. In addition, as the authors suggested, phi is 115 degs for x=1.6 and this is off from 90 degs. But 115 degs is still closer to 90 degs rather than 180 degs. Then, why the four-fold phase is destabilized compared to the two-fold phase? Also, in the authors' discussion, only L-blocks are considered. Why the S- blocks can be neglected to discuss the stability of each phase? Is it possible to estimate angles between μ_L and μ_S in each of phases from magnetization and discuss relative stability between two-fold and four-fold phases? Are there any changes in magnetic anisotropy upon doping Sr and can it be neglected to discuss stability of these magnetic phases? Whatever the case, more straightforward explanation is necessary.

Response:

Firstly, we apologize for the typo made in the previous manuscript. Here (p11) k_0 should be SMALLER instead of LARGER than the k_1 and k_2 , which is fulfilled in all the doping levels.

Secondly, in previous theoretical and experimental studies (ref. 41), it is assumed that all the magnetic moments lie in the ab -plane, then the L blocks always make an angle of ϕ_0 with their neighbors while the neighboring L and S blocks make an angle of $180^\circ - \phi_0/2$ or $180^\circ + \phi_0/2$ in all the IC helix, four-fold and two-fold phases, as shown in the new Fig. 1d in the revised manuscript. That is to say, their relative orientations will be determined for a specific ϕ_0 . For simplicity, we only discuss the L (or) S blocks. The ϕ_0 in IC can be calculated from the magnetic propagation vector k_0 directly while magnetic propagation vectors k_1 and k_2 cannot be used to calculate the ϕ_1 and ϕ_2 in four-fold and two-fold phases. In all those cases, ϕ is only determined by the exchange energy and Zeeman energy. In the revised manuscript, ϕ_2 is estimated by the magnetization along [100]. We found that ϕ_2 increases significantly with increasing Sr content and has a very similar value with that of ϕ_0 at each doping level (see new Fig. 5a). The strong Sr doping dependent ϕ_2 should be also due to the modification of superexchange interactions by Sr doping.

Thirdly, according to previous study, the magnetic anisotropy of S and L block is directly correlated to the Mg site occupation which will be reflected in the saturation magnetization in the collinear ferrimagnetic phase. We have checked the saturation magnetization at 50 kOe for every sample. All the $M_{50\text{ kOe}}$ values are very close to 8 μ_B per f.u. and the maximum deviation is less than 6%, which can be regarded as an error due to the measurements of the sample mass (see Supplementary Information Fig. S3). Therefore, there is no significant change in Mg occupation and magnetic anisotropy with increasing Sr content. As we explained in the revised main text, Sr doping mainly decreases the c -axis lattice constant (see Supplementary Information Fig. S2) and increases the bond angles of Fe(4)-O(2)-Fe(5) and Fe(4)-O(6)-Fe(8) (within the error bar, we did not observe clear trend by single crystal x-ray data since x-ray is not sensitive to the oxygen), as shown in new Fig. 1a. This will directly enhance the exchange interaction between L and S blocks and change the stability of the magnetic phases, consistent with previous theoretical analysis for other Y-type hexaferrites in Ref. 41 and 42. The change of exchange interactions by Sr doping is implied by the significant increase of magnetic transition temperature T_2 from the low-temperature proper screw phase to high-temperature ferromagnetic phase, as indicated by the arrows in Fig. 1e. We have included more discussions in the revised manuscript.

2. Related to above mentioned comments, the authors pointed out that the angle between neighboring μ_L and μ_S changes in two-fold magnetic phase upon increasing x so that P gets larger. Are there any proofs for this change in angles other than change in P ? For example, change in magnetization in two-fold phase upon doping Sr will be useful to estimate such changes in angles and thus should be discussed. P for $x=1.6$ is twice as large as that for $x=0.5$ in two-fold phase, while the

magnetization values in two-fold phase in these two compounds are not so different. In my opinion, the difference in P is too large to be ascribed to the difference in the angle between neighboring μ -L and μ -S. Is it really reasonable?

Response:

First of all, we apologize for the typo related to this comment. The angle mentioned here should be the one between neighboring L and L ($=\phi_2$, defined in the inset of Fig. 1d) and the cited reference should be Ref. 36. According to Ref. 36, $P \sim m_S m_L \sin^2(\phi_2/2)$ for the two-fold spin model, where the $\phi_2/2 = 180^\circ \times \arccos(M_{5\text{kOe}}/M_{50\text{kOe}})$ was estimated from the magnetization data at 5 kOe for two-fold magnetic order and 50 kOe for collinear order, as shown in the added Supplementary Fig. S3 and Fig. 5a. From the equation in Ref. 36, the estimated P for $x = 1.6$ is 1.82 times larger than that for $x = 0$ in two-fold phase. As stimulated by the above comment, we now realize that the extra enhancement of polarization may be related to the enhanced superexchange interaction at the boundary between L and S , because the spin-current or inverse D-M interaction generated P is also proportional to superexchange interaction and the spin-orbit interaction according to Ref. 12. In Y-type hexaferrites, the inverse D-M mechanism induced P is mainly generated by the noncollinear spins at the boundary region between L and S , especially, the Fe(4), Fe(5) and Fe(8) layers (Ref. 40), which is highly frustrated and the exchange interaction can be easily influenced by Ba/Sr doping. These two factors together should be able to account for the increase of polarization with Sr content.

3. It is impossible to judge whether E-induced M-reversal is accompanied by P-reversal or not only from the M-E curve. Therefore, P-E curve for $x=1.6$ should also be provided, if possible.

Response:

We have performed P - H measurements after the slow application of different electric field history to prove the P -reversal after E -induced M -reversal. The results are shown in new Fig. 3b in the revised manuscript. As explained in the revised manuscript, we poled the sample under the condition of $+E$ & $+H$ with the $+E = +2.5$ MV/m to zero H . Then we either remove $+E$ or slowly change the E field to -2.5 MV/m and remove E again to reach either state **1** or **2**, respectively. We subsequently sweep the H to 5 kOe or -5 kOe to measure the magnetoelectric currents and calculate the remnant P value to be about ± 100 $\mu\text{C}/\text{m}^2$ at zero H since the P values of ± 5 kOe do not change under E field (not shown here). From new Fig. 3b, we can clearly see that the state **1** has $+P$ and $+M$ while state **2** has the $-P$ and $-M$, proving that the reversal of M is accompanied with the reversal of P by E field, consistent with the observation in Ref. 33. We have included more discussion related to new Fig. 3b in the revised manuscript.

4. M-T curve for $x=1.6$ (Fig.1e) shows anomaly around 100-130 K. What is the origin of this?

Response:

The reason we measured M - T curve in $H // c$ configuration is to show the low-temperature LC phase with large M value and magnetic phase transitions reported

by other studies for $x = 0$ sample. As suggested by reviewer #4 as well, we have marked in new Fig. 1e the magnetic phase transitions from LC to proper screw (T_1) and from proper screw to the collinear ferrimagnetic (T_2) with black triangles and grey arrows, respectively. For $x = 0$, the $M-T$ profile is consistent with previous study in Ref. 29. However, the $x = 1.6$ shows extra anomaly which may indicate a magnetic phase different from proper screw between 100 - 150 K. So far, we could not determine the exact nature of this phase transition which is beyond the scope of this paper. Further studies are needed in the future.

5. In line 8, page 35, references 33-35 are cited for the previous works on E-induced reversal of M. However, ref. 33 does not include such results at all. Instead, ref. 29 should be cited here.

Response:

We thank the reviewer for pointing out our mistakes and have corrected the cited references in the revised manuscript.

6. Samples measured here are grown by the flux method. Then, the composition of grown crystal, especially the x value can be different from the starting compositions. How do the authors determine x value for each composition?

Response:

Stimulated by this comment, we tried to determine the actual Sr level based on single crystal X-ray diffraction data at room temperature. Due to the role of balancing charge and limited volume in tetrahedral sites, the most probable sites for Mg would be in the middle of the big block of the Fe(12) and Fe(8) octahedral sites. We refined the occupancy of Mg totally 2 units of Mg atoms restrained to the Fe(12) and Fe(8) sites from single crystal X-ray diffraction data. The refinement results show that the actual Sr/Ba ratio is slightly lower than the nominal level, but is consistently increasing with Sr doping, as shown in Supplementary Table S1. The incremental Sr doping is also demonstrated by the monotonic change in the lattice constant c shown in Supplementary Fig. S2.

7. In page 13 “The magnetic properties were carried out...” should be replaced by, for example “Magnetization measurements were carried out...”

Response:

We have revised the sentence mentioned.

8. In the method section, typical dimensions of the samples for each measurement should be provided.

Response:

We have added the sample dimension information in the method section.

Reviewer #2 (Remarks to the Author):

The Authors claim that “This work discloses an effective route to enhance the magnetoelectric effects by tuning magnetic symmetry.” In the paper the way to tune magnetic properties of multiferroic $\text{Sr}_2\text{Mg}_2\text{Fe}_{12}\text{O}_{22}$ is a change of its chemical composition e.g. substitute part of Sr ions by Ba ions. Multiferroic materials exhibit coupled magnetic and ferroelectric properties. Therefore the method of changing their magnetic properties and structures with their chemical composition is a common effect observed and reported in many papers. Composition changes, were widely used in ABO_3 multiferroics to improve their magnetoelectric properties. It is widely accepted that the Dzyaloshinskii-Moriya (D-M) interaction is sensitive to chemical composition of the compound. Therefore the effect presented in the paper was already known. In multiferroics atomic composition change causes a change of their crystal and magnetic structure. This is a consequence also of the Dzyaloshinskii-Moriya interaction present in these materials. Therefore the last sentence in the abstract (lines 31 and 32) “This work discloses an effective route to enhance the magnetoelectric effects by tuning magnetic symmetry” suggests that this tuning was shown the first time by the Authors. I propose to withdraw this sentence from the abstract, or precisely describe what a new idea of improving the magnetoelectric properties of electromagnets is presented in the paper.

As hexaferrites are paraelectric in the absence of external magnetic fields what limits their application in a technique. The single crystal of hexaferrite, $\text{Ba}_{0.4}\text{Sr}_{1.6}\text{Mg}_2\text{Fe}_{12}\text{O}_{22}$, investigated in the paper shows a big magnetoelectric coefficient and a large electric-field-reversed magnetization $\Delta M = 5.3 \mu\text{B}/\text{f.u.}$ It is very interesting and perspective result. This result justifying publication of the paper after its solid reconstructions.

We would like to thank the reviewer for his/her quite positive judgment on our paper: “It is very interesting and perspective result. This result justifying publication of the paper after its solid reconstructions”. We have made corresponding changes in response to the reviewer’s comments as listed below.

The Authors report the changes of magnetic structure of compounds as a function of x (Ba/Sr) ratio via neutron scattering and using magnetization, ferroelectric, and neutron diffraction measurements (for four solid solutions).

Line 61-66 “The study on the Y-type hexaferrite $\text{Ba}_{2-x}\text{Sr}_x\text{Mg}_2\text{Fe}_{12}\text{O}_{22}$ family and achieved giant ME coefficients of $\alpha_H = 33000 \text{ psm-1}$ and $\alpha_E = 32000 \text{ psm-1}$ in $\text{Ba}_{0.4}\text{Sr}_{1.6}\text{Mg}_2\text{Fe}_{12}\text{O}_{22}$, is a new record in single-phase multiferroics”. By exploring how the magnetic phase and ME properties vary with the content of Sr in $\text{Ba}_{2-x}\text{Sr}_x\text{Mg}_2\text{Fe}_{12}\text{O}_{22}$, the Authors claim that they observe the correlation between the giant ME effects and the conical spin structure. This fact has to be better described and documented (see below). It has to be compared with papers of the results of other groups working with these class of materials (e.g. reference [30]).

Paper needs with following changes:

- Withdraw last sentence in the abstract,
- In the Introduction refer to the papers investigating similar effects in other multiferroics,
- D-M and magnetic interactions should be described for Y-type hexaferrites.
- The crystal and magnetic structures and their changes for different Ba/Sr ratio has to be presented in the paper (values and their errors). For at least one solid solution e.g. for Ba_{0.4}Sr_{1.6}Mg₂Fe₁₂O₂₂ the complete structure analysis (also reliability factors R) has to be presented.

Response:

1) Doping is a common way to tune many system including multiferroics. However changing the magnetic symmetry in the way we described in the manuscript is the first time to be revealed by our study to significantly increase the ME coefficients. The specific way can vary from doping, pressure/strain, or other techniques, but the transverse 2-fold magnetic symmetry is critical. To clarify the misleading, we have rewritten the last sentence in the Abstract. We have restricted the conclusive remarks to multiferroic hexaferrites and also emphasize tuning the magnetic symmetry is an effective route by using “tuning the magnetic symmetry” to lead the whole sentence in order to avoid misleading our claim.

2) We have included more description about similar ME effects in other multiferroics in the introduction.

3) We have included description on the D-M and magnetic interactions in Y-type hexaferrites, and discussed the role of D-M interaction and superexchange in the enhancement of electric polarization. Note that the D-M interaction is a very weak term in the free energy of hexaferrites to determine their magnetic structure at zero H or under finite H. Previous theoretical studies usually don't include this term to discuss the stability of field dependent magnetic phases.

4) We have added single crystal X-ray diffraction results measured at room temperature for all the samples in the Supplementary Information. The complete crystal structure for $x=1.6$ refined with our $T=5$ K neutron scattering data is presented in Table S2 in the Supplementary Information. The R factor is less than 7%, proving the reliability of the analysis.

'Block' structure is widely used in explaining magnetic structures of hexaferrites due to their structure complexity. The spins within each block are expected to be aligned collinearly in conical phases, and thus form the 'block-type' conical magnetic structures, as schematically shown in new Fig. 1b and 1c. This schematic picture works well in explaining the multiferroics in hexaferrites. Detailed spin structures for large unit cell system like this is technically very difficult.

Reviewer #3 (Remarks to the Author):

The present manuscript focusses on Sr-doping of the Y-type hexaferrite Ba₂Mg₂Fe₁₂O₂₂ (BFMO) and on its effects on its dielectric and magneto-electric properties. There is a vast amount of literature on Y-type hexaferrite with various cation compositions and doping, but the general understanding of the properties of these materials is poor, partly due to the complexity of the magnetic structures at the microscopic level (beyond the 'block' model). Unquestionably, this paper provides an original contribution to the field, particularly in that the Sr/Ba substitution, although a very natural one, has not been explored in detail. The paper contains much valuable information about the phase diagram, which will be of interest for the researchers that are focusing on these compounds. In this report, I therefore focus on the elements that would make this paper publishable with the high degree of priority and exposure to a wider audience that is granted by Nature Comm. Based on the abstract, this claim is largely based on the giant magneto electric effect and on the large electric-field-reversed magnetisation.

1. Giant magneto-electric effect: there is a plethora of 'giant' effects at low temperatures (e.g., 'Colossal magnetoresistance' in some manganites), and very few of them are of any technological interest whatever. These effects would, of course, be interesting if they denoted new physics, but this does not seem to be the case here. The so-called 'giant magneto-electric effect' here is simply a consequence of the fact that the transverse conical phase exists in zero field for $x = 1.6$. This cone is very easily manipulated so one sees a large change in P (equal to $2P_{\max}$) over a narrow magnetic field interval - hence a large ME effect. In other words, the $x = 1.6$ is simply a continuation of the phase existing at high doping, but 'compressed' towards the zero-field axis. It is not even clear that this phase even displays the linear magneto-electric effect by symmetry,

Response:

Firstly, we call the observed ME effects “giant” because both the direct and converse ME effects represent new records in single phase multiferroic materials, as acknowledged by reviewer #1. The term “giant” or “colossal” was already used in previous publications to describe large ME effects, for example, Chun, S. *et al.* Realization of giant magnetoelectricity in helimagnets. *Phys. Rev. Lett.* 104, 037204 (2010); Oh, Y. S. *et al.* Non-hysteretic colossal magnetoelectricity in a collinear antiferromagnet. *Nat. Commun.* 5, 3201 (2014).

Secondly, multiferroic hexaferrites are among the most promising single-phase materials toward applications because the largest ME effects reported so far at various temperatures are always obtained in hexaferrites. The unprecedentedly high ME effects reported here and the understanding of their origin will stimulate the process of technological applications for single-phase multiferroics.

Thirdly, there is also new physics behind the observed giant ME effects. The reason that the transverse cone can be easily manipulated and reversed over a narrow magnetic field interval is not simple in physics. Only the two-fold transverse cone phase, not other phases, i.e., four-fold transverse cone, has the easy-plane magnetic anisotropy which effectively reduces the required reversal H . As we revealed in the

added theoretical analysis, the stability of the two-fold transverse cone phase and the enhancement of the electric polarization is not straightforward but requires the fine tuning of the lattice structure and strength of various magnetic interactions. And, the two-fold transverse cone for $x = 1.6$ is definitely not a linear magnetoelectric since it shows both the remnant M and P at zero H and zero E (new Fig. 3).

2. The electric-field-reversed magnetisation is, I think more interesting, but the explanation of why this effect occurs is extremely confusing. In general, at these low temperatures there is a huge amount of irreversibility, with many phases coexisting due to either metastability or kinetic trapping (this is clearly visible in the M - H loops). Under these circumstances, it is extremely difficult to understand what is going on, and it may not even be clear which phases are truly polar.

Response:

We find that the confusion in understanding the large E -reversed M may come from the misunderstanding of the M - H loops in Fig. 2a, especially for the $x = 1.6$ sample. Referee #4 also pointed out that the initial magnetization curve is “difficult to distinguish from the other curves in the plot”. The so-called “huge amount of irreversibility, with many phases coexisting” situation is only true for $x < 1.6$ sample while for $x = 1.6$ sample there is negligible hysteresis and coercivity (its coercive field is merely about 8 Oe). Therefore, the huge E -reversed M in $x = 1.6$ can be easily understood by soft transverse cone state with minimal in-plane magnetic anisotropy, as explained in a similar case in Ref. 33. Therefore, following the suggestion of referee #4, we have re-plotted all the initial magnetization curves into open circles to distinguish them from other normal hysteresis M - H curves. Moreover, from the symmetry analysis, it is well known that all the transverse cone phases are polar as this kind of spin configuration can break the space inversion symmetry and induce a polar axis perpendicular to the cone direction and c -axis.

The paper is not particularly well written, and the non-specialist reader would clearly be confused by the continuous references to previous findings (some of them rather dubious) in a very specialised subfield. I have no doubt that the paper could be greatly improved, but I am quite convinced that no improvement could raise to the required standard. I therefore recommend against publication in Nature Comm.

Response:

As we emphasized above, multiferroic hexaferrites are among the most promising single-phase multiferroic materials discovered so far. Although there could be certain difficulty for non-specialist readers to fully understand the physics due to the complex structure and magnetic phases involved, the unprecedentedly high ME effects reported here represent big progress and will have a high impact on the field of multiferroics.

We would like to thank the reviewer for the comments/critics that push us to present the finding and discuss the physics in a better way. Overall, we have made great efforts to improve the manuscript by incorporating all the reviewers' comments/critics,

performing extra experiments, expanding in-depth theoretical analysis, redrawing the essential figures and reorganize the infrastructure of the paper. We expect that the substantially revised paper has met the required standard of Nature Communications and are looking forward to a refreshed view of this reviewer.

Reviewer #4 (Remarks to the Author):

The authors have presented an intriguing result in regards to extremely large magnetoelectric effects in the family of Y-type hexaferrite $Ba_{2-x}Sr_xMg_2Fe_{12}O_{22}$. By studying a range of Sr doping levels, and utilizing a number of characterization techniques, the authors are able to develop a model explaining the large effect observed for $x=1.6$. While an intriguing result worthy of publication, the authors need to address a number of issues in regards to the presentation of their results before they will be suitable for publication. These issues are listed below.

We appreciate the reviewer's great efforts in carefully reviewing our manuscript. All the suggestions are very helpful for us to improve the quality of our manuscript. The responses to all the questions/comments are listed below.

1) Structure of the Y-type hexaferrite $Ba_{2-x}Sr_xMg_2Fe_{12}O_{22}$

For the general reader, the authors should provide a bit more information on the structure of these hexaferrites, for example an explanation of the notation of Y-type, vs. M-, Z-, and U-types. In the crystal structure schematic in Fig. 1(a), the authors should include labels for the in-plane and out-of-plane directions. The yellow atoms may not reproduce well. The authors should give an explanation of the notation of Fe(5) and Fe(4) used in Fig. 1(a). From their figure, it is not possible to visualize the octahedral and tetrahedral coordination of the Fe^{3+}/Mg^{2+} ions. Is that an important factor in this work? What determines the magnitude of the Large and Small magnetic moment layers? On line 102 on the top of page 6, it is not clear what is meant by "calibrating" the neutron and x-ray diffraction data. The two measurements are taken at different temperatures, so how are thermal expansion coefficients taken into account? Are there any structural transitions that occur as a function of temperature between room temperature and 5 K that would need to be taken into account?

Response:

Firstly, we have re-plotted Fig. 1a to show the more detailed structure information and related structure, magnetic and atomic notations. The crystal orientations have also been labeled. The color scheme of each atoms is refined. We also include brief introduction of structure for other hexaferrites. Please refer to the revised manuscript for detailed modification. The octahedral and tetrahedral coordination of the Fe/Mg are plotted in distinguishable colors. The notation and visualization are chosen to be consistent with the review paper by Kimura (Ref.18).

Secondly, the magnitudes of the large and small magnetic moment layers are mainly determined by the Mg occupation since Mg has no net moments and the relative

orientation of each spin within one kind of magnetic blocks is well known. Kimura et al defined in $Ba_{2-x}Sr_xZn_2Fe_{12}O_{22}$ that the effective moment of the Large and Small magnetic moments: $\mu_L=(3+2\gamma) \mu_{Fe}$ and $\mu_s=(2\gamma-1) \mu_{Fe}$, where μ_{Fe} being the moment per Fe site and γ is the portion of Zn occupancy ratio in the L block. While different from Zn, which is well agreed to only occupy the tetrahedral sites, Mg tends to distribute in octahedral Fe sites.

Thirdly, we have changed “calibrated” to “refined”.

Fourthly, no structural transition occurs below 700 K for this hexaferrite system, (our neutron measurement did not observe the structural transition between 1.5 K and 450 K). Both the susceptibility and the neutron measurements did not show a transition or any observable changes between 1.5 K and 10 K. So we think the thermal effect has negligible influence on the structural properties reported here.

2) Magnetolectric phase diagram of $Ba_2Mg_2Fe_{12}O_{22}$

In Fig. 1(d), the authors present their data on the magnetolectric phase diagram of $Ba_2Mg_2Fe_{12}O_{22}$ and denote three regions where the material exists in different magnetic structures. It is not clear whether the data points in green, red, and blue represent experimental points which delineate the transition between these three regimes, or are these data points within each regime? The insets within the panel are a bit small and difficult to read. What is represented by the red arrow?

Response:

We have modified Figs. 1b-1d entirely to clarify the magnetic spin configurations of each phases in Fig. 1d. In the new Fig. 1d, the data points in blacks represent experimental points which delineate the transition between these regimes. The 3D spin configurations of each magnetic structure are plotted in Figs. 1b and 1c first. Then the insets within the panel only draw their in-plane components with larger font size. Each spin structures are drawn in different colors. The red arrow, (which is now blue) indicates that the IC phase lies in the left low H region.

3) Magnetic properties of Y-type hexaferrite $Ba_{2-x}Sr_xMg_2Fe_{12}O_{22}$

The authors present the M vs T curves of their samples in Fig. 1(e). On line 105 on the top of page 6, the authors should highlight that the transition into the proper screw spin structure corresponds to the decrease in magnetization seen below 195 K. The authors neglect to discuss the composition dependence of several features in their M vs T curves, for example, the transition temperature into the proper screw spin structure, as well as the transition temperature into the LC state, and the value of the plateau in magnetization reached at low temperature. Furthermore, the shape of the M vs T curves differs in the $x=1.6$ sample compared to the other compositions. What is the significance/origin of this feature which appears between 100 and 150 K? The authors should indicate what is meant by the FC notation in Fig. 1(e). No y-axis values are listed for Fig. 2(a). An explanation should be given for the lack of values, or the same units as used in Fig. 1(e) and Fig. 3 should be used.

The figure caption to Fig. 2 should explain that the dotted line represents the initial magnetization curve. It is noted, however, that this curve is difficult to distinguish

from the other curves in the plot. The authors state that the hysteresis loops shown in Fig. 2(a) have three discontinuous transitions for the $x=0$ composition, however, this statement ignores the gradual increase in magnetization which occurs for $H < -1.2$ kOe, such that the material does not reach full saturation until above -2 kOe. This feature is also prevalent in the $x=0.5$ sample, though it seem to disappear for $x=1$ and 1.6 . Some discussion should be made for these features. Do the sharp transitions seen in Fig. 2(a) correspond well to the field values of the transitions Fig. 2(b)? The authors should give an explanation of why the H and E fields are applied along the given crystallographic directions. The authors should add a reference to Fig. 2(d) in the text at the bottom of page 7.

Response:

Firstly, we have modified Fig. 1e to indicate the transition temperature into the proper screw spin structure (T_2), as well as the transition temperature into the LC state (T_1). All the Sr substituted samples keep the same trend in $M-T$ curve with a plateau of M with gradually increased transition T_1 at low temperature, indicating the existence of the longitudinal conical state. The T_2 also systematically increase with Sr doping, even though the T_2 for the highest doping is too high to be fully observed. The value of the plateau in magnetization reached at low temperature does not change systematically with Sr doping, which may be attributed to the difference in coercive field. However, we do find influence of Sr doping on T_2 which heavily relies on the energy scale of superexchange interaction between L and S blocks. We have added more discussion on this issue in the main text. As for the extra anomaly in $x = 1.6$, it may indicate a magnetic phase different from proper screw between 100 and 150 K. So far, we could not determine the exact nature of this phase which is beyond the scope of this paper. More studies are needed in the future.

Secondly, we have changed the FC notation in Fig. 1(e) into field-cooling. For Fig. 2, units have been added in Fig. 2a and the initial $M-H$ curves are drawn in open circles to distinguish them from the other curves in the plot. As shown in the figures in Ref. 21, the gradual increase between -1.2 kOe and -2 kOe region corresponds to the phase transition region from four-fold transverse cone to two-fold transverse cone phases, where the spin cone angle changes gradually with increasing magnetic field. The sharp transitions seen in Fig. 2(a) correspond well to the field values of the transitions in Fig. 2(b) for every doping level.

Thirdly, the $H-E$ orientation relationship in the inset of Fig. 2b is due to the fact that the polarization generated by transverse cone from inverse D-M interaction always has the $P \perp H \perp c$ relationship. We have explained this in the revised manuscript. We have added a reference to Fig. 2(d) in the text at the bottom of page 7, as suggested by referee.

4) Overall paper structure

The authors have not maintained a consistent choice of color to represent a given sample throughout all the figures in the manuscript. For example, in Fig. 1(e), the colors are black, red, blue, and pink for the $x=0$, 0.5 , 1.0 , and 1.6 samples, respectively, while the colors are red, green, blue, and dark yellow for Fig. 2. A

different color scheme is used in the supplemental material. The composition of BMFO is given correctly on line 227 on page 11. The authors should give some actual dimensions of the samples in the methods section, since the sample dimensions affect the voltage values needed to achieve a given E field. On line 280 on page 14, “monochromatic” is misspelled. On line 285 of page 14, the Miller index notation is given using the notation for a direction, rather than a plane.

Response:

The colors have been unified for the different doping levels. The composition of BMFO is corrected on line 227 on page 11. The actual dimensions of the samples used in this paper are given in the revised methods section. The typos have been also corrected.

Summary of changes made

1. The last sentence in Abstract is changed.
2. On page 3-4, the introduction is modified to include more information on similar studies, the structural details and classification of all types of hexaferrites.
3. On page 5-7, the subsection “Structural and magnetic properties of the $\text{Ba}_{2-x}\text{Sr}_x\text{Mg}_2\text{Fe}_{12}\text{O}_{22}$ family” has been fully revised to convey the information in a more clear and consistent way.
4. On page 8-9, more descriptions have been included in the subsection “Spin-driven ferroelectricity and giant ME effects”.
5. On page 10, a paragraph has been included to describe that the reversal of M by electric field is accompanied with a reversal of P .
6. On page 13-15, the majority of the discussion section has been revised.
7. More information has been included in the Methods section.
8. More references have been included and the sequence of some references has been adjusted.
9. Figure 1 has been re-plotted with a new figure caption.
10. Figure 2a has been slightly modified.
11. Figure 3 has been replaced with a new Fig. 3a and 3b.
12. Figure 5 has been slightly modified.
13. In Supplementary Information, Fig. S3 and Table S1 and S2 have been included.

All the changes are highlighted in red in the revised manuscript.

Reviewers' comments:

Reviewer #1 (Remarks to the Author):

I have read through authors' rebuttal letter and revised manuscript. The authors revised manuscript based on some additional experiments and analyses. As a result, explanations for why four-fold phase is destabilized and why P increase upon Sr doping get much clearer. In addition, now it is obvious that E-induced M-reversal is actually accompanied by P-reversal. I feel the authors have considered my comments carefully and provided satisfactory responses and modifications. As is originally commented in the previous report, as far as I know, their data are the world records of the ME coefficients in the single phase multiferroic materials and thus, in my opinion, will attract broad and vital interest of researchers.

Therefore, now I can recommend publication of this paper in Nature Communications.

Minor comments:

1. In page 5, 'k' in the KNB formula should be defined.
2. In page 14, line 6, '22' should be subscript.
3. In page 14, jargon such as 'four-fold in-plane phase' must be avoided.

Reviewer #2 (Remarks to the Author):

Manuscript NCOMMS-17-00859 (revised)

Giant magnetoelectric effects achieved by tuning spin cone symmetry in Y-type hexaferrites by Kun Zhai et al. (revised)

The Authors (following my suggestion) changed the last sentence in the abstract and claim that "Our study reveals that tuning magnetic symmetry is an effective route to enhance the magnetoelectric effects in multiferroic hexaferrites".

As the tuning magnetic symmetry (by doping) to enhance the magnetoelectric effects was discovered earlier in many other magnetoelectric substances, I would suggest to add the word "also" into this phrase. Therefore the last sentence in the abstract would be:

"Our study reveals that tuning magnetic symmetry is an effective route to enhance the magnetoelectric effects also in multiferroic hexaferrites".

The single crystal of hexaferrite, $\text{Ba}_{0.4}\text{Sr}_{1.6}\text{Mg}_2\text{Fe}_{12}\text{O}_{22}$, investigated by the Authors, shows a big magnetoelectric coefficient and a large electric-field-reversed magnetization, $\Delta M = 5.3 \mu\text{B}/\text{f.u.}$ It is very interesting and perspective result of the paper.

I found the paper, after revision, suitable for publication in the journal.

Reviewer #4 (Remarks to the Author):

NCOMMS-17-00859A

Revised manuscript

The authors have clearly put a significant amount of effort into revising their manuscript based on the comments of the reviewers. While the manuscript is easier to grasp for readers who are not experts in

this class of materials, I believe that there are still a few remaining issues that need to be addressed before the manuscript can be accepted for publication.

Figure 1:

I am still confused by the schematic of the crystal structure of the Y-type hexaferrite, specifically how the left and right images correlate to one another. After looking at the figure for a while, I believe that the left hand side of each schematic represents a different position in the unit cell. Why are atoms 7 and 8 drawn with dotted lines? "Transverse" is misspelled in panel (c). The relationship between the neighboring L and S blocks is described at the top of page 6 in reference to the IC-LC phase, but is never explicitly stated for the four-fold TC or two-fold TC phases. However, in the rebuttal letter, it is my understanding that the same angle relationship holds for all three phases, and it would be instructive if this point was made explicitly in the manuscript. A minor point is that the phase diagram in panel (d) is stated to be divided into three areas below 100 K, and these areas as labeled as [1], [2], and [3], and within these areas, the authors show the in-plane magnetic structures with in-plane turn angles of ϕ_0 , ϕ_1 , and ϕ_2 , respectively. The difference in numbering make it non-intuitive to correlate the regions and their associated turn angle.

Figure 3: I like the new data included in Fig. 3.

Figure 4: When denoting the Bragg diffraction peaks, the proper Miller index notation is without commas. For the $x=1.6$ sample, why do the incommensurate peaks remain at 2 KOe even though the magnetic hysteresis loops show that the sample has been saturated at a significantly smaller field? I suspect that these peaks are significantly smaller in magnitude than the commensurate peaks, but the way the y-axis scale has been truncated, it is not possible to clearly see that that is the case. The authors state that the incommensurate K_0 and the commensurate K_1 peaks for the $x=0.5$ and 1.0 samples are indistinguishable within the resolution of the neutron diffraction measurements. At the bottom of page 11, two values of K_0 are listed, but it is not stated clearly which samples these values correspond to. It may be clarified if they end the sentence with the word "respectively", if that is indeed the case. Can the authors give error bars on the ϕ_0 angles for the incommensurate phase determined from the neutron diffraction data? I think it would be instructive if the authors made correlations between the hysteresis loops (Fig. 2(a)) and the neutron diffraction data. For example, if I understand the sample conditions correctly, the initial $H=0$ kOe scan represents the 'virgin' sample prior to the initial magnetization curve, while the final $H=0$ kOe scan is the sample after the positive branch of the hysteresis loop. For the $x=1.6$ sample, the incommensurate peaks are not observed in the final $H=0$ kOe case, which indicates that the commensurate two-fold phase is stable at $H=0$ kOe, which is also the case observed in the hysteresis loop. In the other samples, the remanent magnetization is nearly zero, and the incommensurate peaks return, though the shape and intensity of the peaks differ. The authors could make a comment to denote what structural differences could exist between the virgin and remanent states.

Figure 5: In panel (c), it is not clear to me why each experimental data point is represented with three symbols for the M-H, P-H, and epsilon-H measurements as they all seem to be lie on top of one another.

Supplemental material: The authors state that the Sr doping mainly affects the c-axis lattice parameter, but it would be helpful if the authors showed a comparison with the a- and b-axis lattice parameter.

Composition of samples: On the top of page 7, the authors state that the "Sr/Ba ratio is slightly lower", lower than what quantity? The nominal doping level? How do the compositions of the other elements compare to their nominal values?

M vs. T curves: The authors refer repeatedly to a “plateau” region in the M vs. T curves, but it is not clear to me which region this refers to as there are three main regions with nearly constant magnetization. The authors could instead refer to this region as the plateau region below/above T1/T2, whichever range is appropriate in this case.

Discussion: The discussion on pages 14 and 15 are difficult to follow, as they are written for a reader with a strong understanding of the Y-type hexaferrite phase. This section should be re-written to increase the understanding as it is a vital portion of the manuscript. I believe that in part of the discussion on page 15, the authors are trying to state that the superexchange interactions are increasing with increasing x value. Earlier in their paper they state that the increase in T2 value with increasing x is an experimental proof that the superexchange interaction increases, and this point should be reiterated in the discussion section.

Responses to the reviewers' comments

Reviewer #1 (Remarks to the Author):

Minor comments:

1. In page 5, 'K' in the KNB formula should be defined.

Reply: We have defined “K” as suggested.

2. In page 14, line 6, '22' should be subscript.

Reply: We have corrected the mistake.

3. In page 14, jargon such as ‘four-fold in-plane phase’ must be avoided.

Reply: We have modified the entire discussion section to make it easier for the readers, as also suggested by the referee #4.

Reviewer #2 (Remarks to the Author):

As the tuning magnetic symmetry (by doping) to enhance the magnetoelectric effects was discovered earlier in many other magnetoelectric substances, I would suggest to add the word “also” into this phrase. Therefore the last sentence in the abstract would be: “Our study reveals that tuning magnetic symmetry is an effective route to enhance the magnetoelectric effects also in multiferroic hexaferrites”.

Reply: We have changed the last sentence as the reviewer suggested.

Reviewer #4 (Remarks to the Author):

The authors have clearly put a significant amount of effort into revising their manuscript based on the comments of the reviewers. While the manuscript is easier to grasp for readers who are not experts in this class of materials, I believe that there are still a few remaining issues that need to be addressed before the manuscript can be accepted for publication.

Figure 1:

I am still confused by the schematic of the crystal structure of the Y-type hexaferrite, specifically how the left and right images correlate to one another. After looking at the figure for a while, I believe that the left hand side of each schematic represents a different position in the unit cell.

Reply: We have modified Fig. 1a and explained in the Figure caption that the left panel is a magnified diagram representing the dotted square region.

Why are atoms 7 and 8 drawn with dotted lines?

Reply: Since atoms 7 and 8 lie outside of the (110) plane, the dotted circles represent the trajectory of those atoms onto the (110) plane. We have added this description in the caption.

“Transverse” is misspelled in panel (c).

Reply: We have corrected it.

The relationship between the neighboring L and S blocks is described at the top of page 6 in reference to the IC-LC phase, but is never explicitly stated for the four-fold TC or two-fold TC phases. However, in the rebuttal letter, it is my understanding that the same angle relationship holds for all three phases, and it would be instructive if this point was made explicitly in the manuscript.

Reply: We have added a sentence “In those phases, the turning angles between neighboring *L* and *S* also follow the same relationship” to explain this point in four-fold and two-fold phases.

A minor point is that the phase diagram in panel (d) is stated to be divided into three areas below 100 K, and these areas are labeled as [1], [2], and [3], and within these areas, the authors show the in-plane magnetic structures with in-plane turn angles of ϕ_0 , ϕ_1 , and ϕ_2 , respectively. The difference in numbering makes it non-intuitive to correlate the regions and their associated turn angle.

Reply: Following the reviewer’s suggestion, the three regions have been relabeled as [0], [1], [2] (see new Fig. 1d).

Figure 4: When denoting the Bragg diffraction peaks, the proper Miller index notation is without commas.

Reply: We have removed the commas in Miller index notation.

For the $x=1.6$ sample, why do the incommensurate peaks remain at 2 kOe even though the magnetic hysteresis loops show that the sample has been saturated at a significantly smaller field? I suspect that these peaks are significantly smaller in magnitude than the commensurate peaks, but the way the y-axis scale has been truncated, it is not possible to clearly see that that is the case.

Reply: We carefully checked the M-H data of $x=1.6$ in Fig. 2a. We found that the 2 kOe is not enough to fully saturate the M, even though the initial curve and the down sweeping curve are very close. Therefore, the small incommensurate peaks at 2 kOe are a consequence of a small portion of incommensurate magnetic phase. For the data with magnetic fields higher than 2.6 kOe and down sweep zero field data, these peaks are definitely not there after checking the data carefully.

The authors state that the incommensurate K0 and the commensurate K1 peaks for the $x=0.5$ and 1.0 samples are indistinguishable within the resolution of the neutron diffraction measurements. At the bottom of page 11, two values of K0 are listed, but it is not stated clearly which samples these values correspond to. It may be clarified if they end the sentence with the word “respectively”, if that is indeed the case. Can the authors give error bars on the ϕ_0 angles for the incommensurate phase determined from the neutron diffraction data?

Reply: Actually, we already included the error bar in Fig. 5a for the calculated ϕ_0 . We have replotted Fig. 5a to clarify the symbol errors. Usually, the error bar of ϕ_0 is about ± 2 degree, which corresponds to ± 0.02 in K values determined from the neutron diffraction data. By the way, we made a typo in the formula of calculating ϕ_0 in the caption of Fig. 5, which has been corrected in the new revision. This typo does not affect the calculated ϕ_0 values.

I think it would be instructive if the authors made correlations between the hysteresis loops (Fig. 2(a)) and the neutron diffraction data. For example, if I understand the sample conditions correctly, the initial $H=0$ kOe scan represents the ‘virgin’ sample prior to the initial magnetization curve, while the final $H=0$ kOe scan is the sample after the positive branch of the hysteresis loop. For the $x=1.6$ sample, the incommensurate peaks are not observed in the final $H=0$ kOe case, which indicates that the commensurate two-fold phase is stable at $H=0$ kOe, which is also the case observed in the hysteresis loop. In the other samples, the remanent magnetization is nearly zero, and the incommensurate peaks return, though the shape and intensity of the peaks differ. The authors could make a comment to denote what structural differences could exist between the virgin and remanent states.

Reply: Simply from the hysteresis loop, we cannot easily determine the magnetic phase of the final $H = 0$ kOe scan. By comparing the initial up-sweep M-H curve and the up-sweep H scan curve, the phase transition fields are different for $x > 0$ samples. Therefore, the only comment we can make is that “detailed initial and final magnetic states at $H = 0$ kOe should be different for $x = 0.5$ and 1.0 samples even though the incommensurate peak positions of those samples return after the removal of external field”. Then the difference in shape and intensity of the peaks in neutron data can be understood. We have added some discussions in the main text.

Figure 5: In panel (c), it is not clear to me why each experimental data point is represented with three symbols for the M-H, P-H, and epsilon-H measurements as they all seem to be lie on top of one another.

Reply: We use different symbols to represent the phase transition field obtained by different measurement technique, square for M-H, circle for P-H and star for e-H. The overlap of the three kinds of symbols simply indicate that different measurements point to the same phase transition processes and the phase diagram we draw is justified.

Supplemental material: The authors state that the Sr doping mainly affects the c-axis lattice parameter, but it would be helpful if the authors showed a comparison with the a- and b-axis lattice parameter.

Reply: We have included a-axis lattice parameter. As the material has a hexagonal symmetry, the b-axis is the same as the a-axis.

Composition of samples: On the top of page 7, the authors state that the “Sr/Ba ratio is slightly lower”, lower than what quantity? The nominal doping level? How do the compositions of the other elements compare to their nominal values?

Reply: Yes, we mean that the refined Sr/Ba ratio is lightly lower than nominal doping level. We have clarified this in the new revision. Due to the large unit cell and the poor resolution of XRD to oxygen, we constrained the atomic ratio as that in the well-accepted chemical formula. The total occupancy of Ba and Sr was fixed to 100%, their ratio was refined and shown in Supplementary Table 1.

M vs. T curves: The authors refer repeatedly to a “plateau” region in the M vs. T curves, but it is not clear to me which region this refers to as there are three main regions with nearly constant magnetization. The authors could instead refer to this region as the plateau region below/above T_1/T_2 , whichever range is appropriate in this case.

Reply: We have modified the description as suggested.

Discussion: The discussion on pages 14 and 15 are difficult to follow, as they are written for a reader with a strong understanding of the Y-type hexaferrite phase. This section should be re-written to increase the understanding as it is a vital portion of the manuscript.

Reply: We are grateful for your suggestion. We had put a great effort to reorganize our discussion part to be more understandable for readers that lack a deep understanding of hexaferrites. As the Reviewer 1# suggested, we are also trying to avoid using jargon such as “four-fold in-plane phase” and use more general physical picture to be easily understood.

I believe that in part of the discussion on page 15, the authors are trying to state that the superexchange interactions are increasing with increasing x value. Earlier in their paper they state that the increase in T_2 value with increasing x is an experimental proof that the superexchange interaction increases, and this point should be reiterated in the discussion section.

Reply: It is a good suggestion. We have restated that “ T_2 value with increasing x might be also an experimental proof that the superexchange interaction increases” in the discussion part.

Reviewers' Comments:

Reviewer #4:

Remarks to the Author:

The authors have done a good job to address my comments. The revised manuscript is appropriate for publication.

REVIEWERS' COMMENTS:

Reviewer #4 (Remarks to the Author):

The authors have done a good job to address my comments. The revised manuscript is appropriate for publication.

Response: we thank the reviewer for the recommendation for publication.